# CAUSAL UNSUPERVISED SEMANTIC SEGMENTATION

## ABSTRACT

Unsupervised semantic segmentation aims to achieve high-quality semantic grouping without human-labeled annotations. With the advent of self-supervised pre-training, various frameworks utilize the pre-trained features to train prediction heads for unsupervised dense prediction. However, a significant challenge in this unsupervised setup is determining the appropriate level of clustering required for segmenting concepts. To address it, we propose a novel framework, CAusal Unsupervised Semantic sEgmentation (CAUSE), which leverages insights from causal inference. Specifically, we bridge intervention-oriented approach (*i.e.*, frontdoor adjustment) to define suitable two-step tasks for unsupervised prediction. The first step involves constructing a concept clusterbook as a mediator, which represents possible concept prototypes at different levels of granularity in a discretized form. Then, the mediator establishes an explicit link to the subsequent concept-wise self-supervised learning for pixel-level grouping. Through extensive experiments and analyses on various datasets, we corroborate the effectiveness of CAUSE and achieve state-of-the-art performance in unsupervised semantic segmentation.

## 1 INTRODUCTION

Semantic segmentation is one of the essential computer vision tasks that has continuously advanced in the last decade with the growth of Deep Neural Networks (DNNs) (He et al., 2016; Dosovitskiy et al., 2020; Carion et al., 2020) and large-scale annotated datasets (Everingham et al., 2010; Cordts et al., 2016; Caesar et al., 2018). However, obtaining such pixel-level annotations for dense prediction requires an enormous amount of human resources and is more time-consuming compared to other image analysis tasks. Alternatively, weakly-supervised semantic segmentation approaches have been proposed to relieve the costs by using of facile forms of supervision such as class labels (Wang et al., 2020b; Zhang et al., 2020a), scribbles (Lin et al., 2016), bounding boxes (Dai et al., 2015; Khoreva et al., 2017), and image-level tags (Xu et al., 2015; Tang et al., 2018).

While relatively few works have been dedicated to explore unsupervised semantic segmentation (USS), several methods have presented the way of segmenting feature representations without any annotated labels by exploiting visual consistency maximization (Ji et al., 2019; Hwang et al., 2019), multi-view equivalence (Cho et al., 2021), or saliency priors (Van Gansbeke et al., 2021; Ke et al., 2022). In parallel with segmentation researches, recent self-supervised learning frameworks (Caron et al., 2021; Bao et al., 2022) using Vision Transformer have observed that their representations exhibit semantic consistency at the pixel-level scale for object targets. Based on such intriguing properties of self-supervised training, recent USS methods (Hamilton et al., 2022; Ziegler & Asano, 2022; Yin et al., 2022; Zadaianchuk et al., 2023; Li et al., 2023; Seong et al., 2023) have employed the pre-trained features as a powerful source of prior knowledge and introduced contrastive learning frameworks by maximizing feature correspondence for the unsupervised segmentation task.

In this paper, we begin with a fundamental question for the unsupervised semantic segmentation: *How can we define what to cluster and how to do so under an unsupervised setting?*, which has been overlooked in previous works. A major challenge for USS lies in the fact that unsupervised segmentation is more akin to clustering rather than semantics with respect to pixel representation. Therefore, even with the support of self-supervised representation, the lack of awareness regarding what and how to cluster for each pixel representation makes USS a challenging task, especially when aiming for the desired level of granularity. For example, elements such as *head, torso, hand, leg, etc.,* should ideally be grouped together under the broader-level category *person*, a task that previous methods (Hamilton et al., 2022; Seong et al., 2023) have had difficulty accomplishing, as in

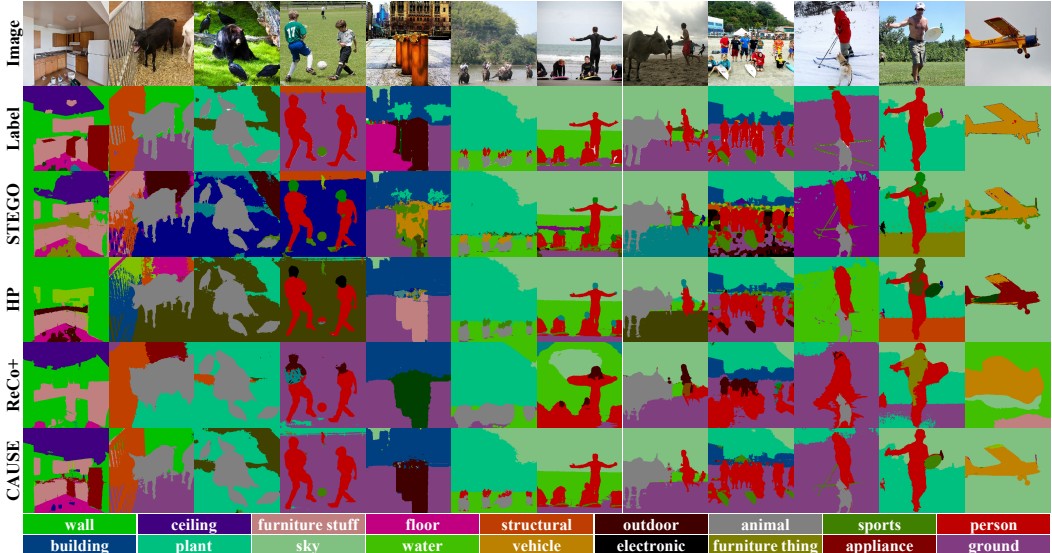

Figure 1: Visual comparison of USS for COCO-stuff (Caesar et al., 2018). Note that, in contrast to true labels, baseline frameworks (Hamilton et al., 2022; Seong et al., 2023; Shin et al., 2022) fail to achieve targeted level of granularity, while CAUSE successfully clusters *person, sports, vehicle*, etc.

Fig. 1. To address these difficulties, we, for the first time, treat USS procedure within the context of causality and propose suitable two-step tasks for the unsupervised learning. As shown in Fig. 2, we first schematize a causal diagram for a simplified understanding of causal relations for the given variables and the corresponding unsupervised tasks for each step. Note that our main goal is to group semantic concepts $Y$ that meet the targeted level of granularity, utilizing feature representation $T$ from pre-trained self-supervised methods such as DINO (Caron et al., 2021).

Specifically, the unsupervised segmentation ($T \rightarrow Y$) is a procedure for deriving semantically clustered groups $Y$ distilled from pre-trained features $T$. However, the indeterminate $U$ of unsupervised prediction (*i.e.,* what and how to cluster) can lead confounding effects during pixel-level clustering without supervision. Such effects can be considered as a backdoor path ($T \leftarrow U \rightarrow Y$) that hinders the targeted level of segmentation. Accordingly, our primary insight stems from constructing a subdivided concept representation $M$, with discretized indices, which serves as an explicit link between $T$ and $Y$ in alternative forms of supervision. Intuitively, the construction of subdivided concept *clusterbook* $M$ implies the creation of as many inherent concept prototypes as possible in advance, spanning various levels of granularity. Subsequently, for the given pre-trained features, we train a segmentation head that can effectively consolidate the concept prototypes into the targeted broader-level categories using the constructed clusterbook. This strategy involves utilizing the discretized indices within $M$ to identify positive and negative features for the given anchor, enabling concept-wise self-supervised learning.

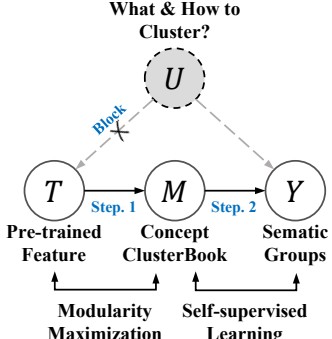

Figure 2: Causal diagram of CAUSE. We split USS into two steps to identify relation between pre-trained features $T$ and semantic groups $Y$ using *clusterbook* $M$.

Beyond the intuitive causal procedure of USS, building a mediator $M$ can be viewed as a blocking procedure of the backdoor paths induced from $U$ by assigning possible concepts in discretized states such as in Van Den Oord et al. (2017); Esser et al. (2021). That is, it satisfies a condition for frontdoor adjustment (Pearl, 1993), which is a powerful causal estimator that can establish only causal association[1] ($T \rightarrow M \rightarrow Y$). We name our novel framework as CAusal Unsupervised Semantic sEgmentation (CAUSE), which integrates the causal approach into the field of USS. As illustrated in Fig. 2, in brief, we divide the unsupervised dense prediction into two step tasks: (1) discrete subdivided representation learning with *Modularity* theory (Newman, 2006) and (2) conducting do-calculus (Pearl, 1995) with self-supervised learning (Oord et al., 2018) in the absence of annotations.

---

[1]In Step 1, $Y$ is a collider variable in the path of $T \rightarrow Y$ through $U$, and it blocks backdoor path. Therefore, causal association only flows into $M$ from $T$. Then, in Step 2, $T$ blocks $M \leftarrow T \leftarrow U \rightarrow Y$. By combining two steps, we can distill the pre-trained representation using only causal association path and reflect it on semantic groups, which is our ultimate goal for unsupervised semantic segmentation. Please see preliminary in Section 3.1.

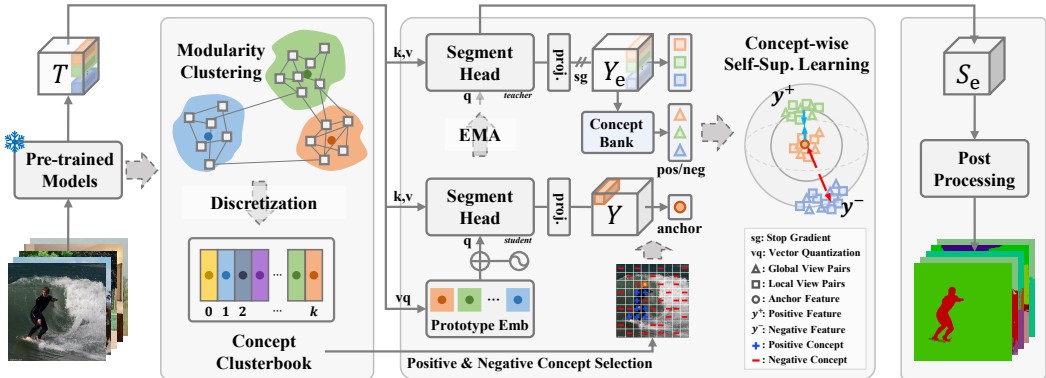

Figure 3: The overall architecture of CAUSE comprises (i): constructing discretized concept cluster-book as a mediator and (ii): clustering semantic groups using concept-wise self-supervised learning.

By combining the above tasks, we can bridge causal inference into the unsupervised segmentation and obtain semantically clustered groups with the support of pre-trained feature representation.

Our main contributions can be concluded as: (i) We approach unsupervised semantic segmentation task with an intervention-oriented approach (*i.e.,* causal inference) and propose a novel unsupervised dense prediction framework called CAusal Unsupervised Semantic sEgmentation (CAUSE), (ii) To address the ambiguity in unsupervised segmentation, we integrate frontdoor adjustment into USS and introduce two-step tasks: deploying a discretized concept clustering and concept-wise self-supervised learning, and (iii) Through extensive experiments, we corroborate the effectiveness of CAUSE on various datasets and achieve state-of-the-art results in unsupervised semantic segmentation.

## 2 RELATED WORK

As an early work for USS, Ji et al. (2019) have proposed IIC to maximize mutual information of feature representations from augmented views. After that, several methods have further improved the segmentation quality by incorporating inductive bias in the form of cross-image correspondences (Hwang et al., 2019; Cho et al., 2021; Wen et al., 2022) or saliency information in an end-to-end manner (Van Gansbeke et al., 2021; Ke et al., 2022). Recently, with the discovery of semantic consistency for pre-trained self-supervised frameworks (Caron et al., 2021), Hamilton et al. (2022) have leveraged the pre-trained features for the unsupervised segmentation. Subsequently, various works (Wen et al., 2022; Yin et al., 2022; Ziegler & Asano, 2022) have utilized the self-supervised representation as a form of pseudo segmentation labels (Zadaianchuk et al., 2023; Li et al., 2023) or a pre-encoded representation to further incorporate additional prior knowledge (Van Gansbeke et al., 2021; Zadaianchuk et al., 2023) into the segmentation frameworks. Our work aligns with previous studies (Hamilton et al., 2022; Seong et al., 2023) in the aspect of refining segmentation features using pre-trained representations without external information. However, we highlight that the lack of a well-defined clustering target in the unsupervised setup leads to suboptimal segmentation quality. Accordingly, we interpret USS within the context of causality, bridging the construction of discretized representation with pixel-level self-supervised learning (see extended explanations in Appendix A.)

## 3 CAUSAL UNSUPERVISED SEMANTIC SEGMENTATION

### 3.1 DATA GENERATING PROCESS FOR UNSUPERVISED SEMANTIC SEGMENTATION

**Preliminary.** It is important to define Data Generating Process (DGP) early in the process for causal inference. DGP outlines the causal relationships between treatment $T$ and outcome of our interest $Y$, and the interrupting factors, so-called confounder $U$. For example, if we want to identify the causal relationship between *smoking* (*i.e.,* treatment) and *lung cancer* (*i.e.,* outcome of our interest), *genotype* can be deduced as one of potential confounders that provoke confounding effects between *smoking* $T$ and *lung cancer* $Y$. Once we define the confounder $U$, and if it is observable, backdoor adjustment (Pearl, 1993) is an appropriate solution to estimate the causal influence between $T$ and $Y$ by controlling $U$. However, not only in the above example but also in many real-world scenarios, including high-dimensional complex DNNs, confounder is often unobservable and either uncontrollable. In this condition, controlling $U$ may not be a feasible option, and it prevents us from precisely establishing the causal relationship between $T$ and $Y$.

---

**Algorithm 1** (STEP 1) Maximizing Modularity for Constructing Concept Clusterbook $M$

---

**Require:** Image Samples $X \sim$ Data, Pre-trained Model $f$, Concept Fractions $M \in \mathbb{R}^{k \times c}$
1: Initialize $M$
2: **for** $X \sim$ Data **do**
3:     $T \in \mathbb{R}^{hw \times c} \leftarrow f(X)$                              ▷ Pre-trained Model Representation
4:     $\mathcal{A} \leftarrow \max(0, \cos(T, T)) \in \mathbb{R}^{hw \times hw}$                      ▷ Affinity matrix
5:     $d, e \leftarrow \mathcal{A}$                   ▷ Degree Matrix and Number of Total edges
6:     $\mathcal{C} \leftarrow \max(0, \cos(T, M)) \in \mathbb{R}^{hw \times k}$                ▷ Cluster Assignment Matrix
7:     $\mathcal{H} \leftarrow \frac{1}{2e}\text{Tr}\left(\tanh\left(\frac{\mathcal{C}\mathcal{C}^T}{\tau}\right)\left[\mathcal{A} - \frac{dd^T}{2e}\right]\right)$        ▷ Maximizing Modularity ($\tau = 0.1$)
8:     $M \leftarrow \text{Increase}(\mathcal{H})$           ▷ Updating Concept ClusterBook (lr: 0.001)
9: **end for**

---

Fortunately, Pearl (2009) introduces frontdoor adjustment allowing us to elucidate the causal association even in the presence of unobservable confounder $U$. Here, the key successful points for frontdoor adjustment are two factors, as shown in Fig. 2: (a) assigning a mediator $M$ bridging treatment $T$ into outcome of our interest $Y$ while being independent with confounder $U$ and (b) averaging all possible treatments between the mediator and outcome. When revisiting the above example, we can instantiate a mediator $M$ as accumulation of *tar* in lungs, which only affects *lung cancer* $Y$ from *smoking* $T$. We then average the probable effect between *tar* $M$ and *lung cancer* $Y$ across all of the participants' population $T'$ on *smoking*. The following formulation represents frontdoor adjustment:

$$p(Y \mid \text{do}(T)) = \underbrace{\sum_{m \in M} p(m \mid T)}_{\text{Step 1}} \underbrace{\sum_{t' \in T'} p(Y \mid t', m)p(t')}_{\text{Step 2}}, \tag{1}$$

where $\text{do}(\cdot)$ operator describes do-calculus (Pearl, 1995), which indicates intervention on treatment $T$ to block unassociated backdoor path induced from $U$ between the treatments and outcome of interest.

**Causal Perspective on USS.** Bridging the causal view into unsupervised semantic segmentation, our objective is clustering semantic groups $Y$ with a support of pre-trained self-supervised features $T$. Here, in unsupervised setups, we define $U$ as indetermination during clustering (*i.e.,* a lack of awareness about what and how to cluster), which cannot be observed within the unsupervised context. Therefore, in Step 1 of Eq. (1), we first need to build a mediator directly relying on $T$ while being independent with the unobserved confounder $U$. To do so, we construct concept clusterbook as $M$, which is set of concept prototypes that encompass potential concept candidates spanning different levels of granularity only through $T$. The underlying assumption for the construction of $M$ is based on the object alignment property observed in recent self-supervised methods (Caron et al., 2021; Oquab et al., 2023), a characteristic exploited by Hamilton et al. (2022); Seong et al. (2023). Next, in Step 2 of Eq. (1), we need to determine whether to consolidate or separate the concept prototypes into the targeted semantic-level groups $Y$. We utilize the discretized indices from $M$ for discriminate positive and negative features for the given anchor and conduct concept-wise self-supervised learning. The following is an approximation of Eq. (1) for the unsupervised dense prediction:

$$\mathbb{E}_{t \in T}\left[p(Y \mid \text{do}(t))\right] = \mathbb{E}_{t \in T}\left[\sum_{m \in M} p(m \mid t) \sum_{t' \in T'} p(Y \mid t', m)p(t')\right], \tag{2}$$

where, $T'$ indicates a population of all feature points, but notably in a pixel-level manner suitable for dense prediction. In summary, our focus is enhancing $p(Y|\text{do}(t))$ for feature points $t$ by assigning appropriate unsupervised two tasks (i) $p(m|t)$: construction of concept clusterbook and (ii) $p(Y|t', m)$: concept-wise self-supervised learning, all of which can be bridged to frontdoor adjustment.

## 3.2 CONSTRUCTING CONCEPT CLUSTERBOOK FOR MEDIATOR

**Concept Prototypes.** We initially define a mediator $M$ and maintain it as a link between the pretrained features $T$ and the semantic groups $Y$. This mediator necessitates an explicit representation that transforms the continuous representation found in pre-trained self-supervised frameworks, into a discretized form. One of possible approaches is reconstruction-based vector-quantization (Van Den Oord et al., 2017; Esser et al., 2021) that is well-suited for generative modeling. However, for dense prediction, we require more sophisticated pixel-level clustering methods that consider pixel

---

**Algorithm 2** (STEP 2): Enhancing Likelihood of Semantic Groups through Self-Supervised Learning

---

**Require:** Head $S;\theta_S$, Head-EMA $S_{\text{ema}};\theta_{S_{\text{ema}}}$, Clusterbook $M$, Distance $\mathcal{D}_M$, Concept Bank $Y_{\text{bank}}$

1: **for** $X \sim$ Data **do**
2: $\quad T \leftarrow f(X)$ $\qquad\qquad\qquad\qquad\qquad\qquad\qquad$ ▷ Pre-trained Model Representation
3: $\quad Q \leftarrow T$ $\qquad\qquad\qquad\qquad\qquad\qquad\qquad\qquad$ ▷ Vector Quantization from $M$
4: $\quad Y, Y_{\text{ema}} \leftarrow S(Q,T), S_{\text{ema}}(Q,T)$ $\quad$ (∗ MLP: $S(T), S_{\text{ema}}(T)$) ▷ Segmentation Head Output
5: $\quad y \sim Y$ $\qquad\qquad\qquad\qquad\qquad\qquad\qquad$ ▷ Anchor Selection (Appendix B for Detail)
6: $\quad y^+, y^- \sim \{Y_{\text{ema}}, Y_{\text{bank}} \mid y\}$ ▷ Positive/Negative Selection from $\mathcal{D}_M$ (Appendix B for Detail)
7: $\quad p \leftarrow \mathbb{E}_y \left[ \log \mathbb{E}_{y^+} \left[ \frac{\exp(\cos(y,y^+)/\tau)}{\exp(\cos(y,y^+)/\tau) + \sum_{y^-} \exp(\cos(y,y^-)/\tau)} \right] \right]$ $\quad$ ▷ Self-supervised Learning
8: $\quad \theta_S \leftarrow \text{Increase}(p)$ $\qquad\qquad$ ▷ Updating Parameters of Segmentation Head (lr: 0.001)
9: $\quad \theta_{S_{\text{ema}}} \leftarrow \lambda \theta_{S_{\text{ema}}} + (1-\lambda)\theta_S$ $\qquad\qquad$ ▷ Exponential Moving Average ($\lambda : 0.99$)
10: $\quad Y_{\text{bank}} \leftarrow \mathbf{R}^2(Y_{\text{bank}}, Y_{\text{ema}})$ $\qquad\qquad$ ▷ $\mathbf{R}^2$: **R**andom Cut $Y_{\text{bank}}$ and **R**andom Sample $Y_{\text{ema}}$
11: **end for**

---

locality and connectivity. Importantly, they should be capable of constructing such representations in discretized forms for alternative role of supervisions. Accordingly, we exploit a clustering method that maximizes modularity (Newman, 2006), which is one of the most effective approaches for considering relations among vertices. The following formulation represents maximizing a measure of modularity $\mathcal{H}$ to acquire the discretized concept fractions from pre-trained features $T$:

$$\max_M \mathcal{H} = \frac{1}{2e}\text{Tr}\left( \mathcal{C}(T,M)^{\text{T}} \left[ \mathcal{A}(T) - \frac{dd^{\text{T}}}{2e} \right] \mathcal{C}(T,M) \right) \in \mathbb{R}, \qquad (3)$$

where $\mathcal{C}(T,M) \in \mathbb{R}^{hw \times k}$ denotes cluster assignment matrix such that $\max(0, \cos(T,M))$ between all $hw$ patch feature points in pre-trained features $T \in \mathbb{R}^{hw \times c}$ and all $k$ concept prototypes in $M \in \mathbb{R}^{k \times c}$. The cluster assignment matrix implies how each patch feature point is close to concept prototypes. In addition, $\mathcal{A}(T) \in \mathbb{R}^{hw \times hw}$ indicates the affinity matrix of $T = \{t \in \mathbb{R}^c\}^{hw}$ such that $\mathcal{A}_{ij} = \max(0, \cos(t_i, t_j))$ between the two patch feature points $t_i, t_j$ in $T$, which represents the intensity of connections among vertices. Note that, degree vector $d \in \mathbb{R}^{hw}$ describes the number of the connected edges in its affinity $\mathcal{A}$, and $e \in \mathbb{R}$ denotes the total number of the edges.

By jointly considering cluster assignments $\mathcal{C}(T,M)$ and affinity matrix $\mathcal{A}(T)$ at once, in brief, maximizing modularity $\mathcal{H}$ constructs the discretized concept clusterbook $M$ taking into account the patch-wise locality and connectivity in pre-trained representation $T$. In practical, directly calculating Eq. (3) can lead to much small value of $\mathcal{H}$ due to multiplying tiny elements of $\mathcal{C}$ twice. Thus, we use trace property and hyperbolic tangent with temperature term $\tau$ to scale up $\mathcal{C}$ (see Appendix B). Algorithm 1 provides more details on achieving maximizing modularity to generate concept clusterbook $M$, where we train only one epoch with Adam (Kingma & Ba, 2015) optimizer.

### 3.3 Enhancing Likelihood for Semantic Groups

**Concept-Matched Segmentation Head.** As part of Step 2, to embed segmentation features $Y$ that match with concept prototypes from pre-trained features $T$, we train a task-specific prediction head $S$. As in Fig. 3, the pre-trained model remains frozen, and their features $T = \{t \in \mathbb{R}^c\}^{hw}$ are fed into the segmentation head $S$ that performs cross-attention with querying prototype embedding $Q = \{q \in \mathbb{R}^c\}^{hw}$. Here, for the given patch features $T$, the prototype embedding $Q$ represents a vector-quantized outputs, which indicates the most representative concept $q = \arg\max_{m \in M} \cos(t, m) \in \mathbb{R}^c$ within the concept clusterbook $M$. The segmentation head $S$ comprises a single transformer layer followed by a MLP projection layer only used for training, and we can derive a concept-matched feature $Y = \{y \in \mathbb{R}^r\}^{hw}$ for concept fractions in $M$, satisfying $Y = S(Q,T)$ (refer to Appendix B).

**Concept-wise Self-supervised Learning.** Using the concept-attended segmentation features, we proceed to enhance the likelihood $p(Y|t', m)$ for effectively clustering pixel-level semantics. To easily handle it, we first re-formulate it as $p(Y|t', m) = \prod_{y \in Y} p(y|t', m)^2$, recognizing that $Y$

---

[2] We only utilize the most closest concept at every patch feature point $t$ in $T$. Hence, $p(m|t)$ of Step 1 can be calculated by using sharpening technique: $p(m=q|t)=1$ if it is $q=\arg\max_{m \in M}\cos(m,t)$; otherwise, $p(m|t)=0$. Then, enhancing $\mathbb{E}_{t \in T}[p(Y|\text{do}(t))]$ for our main purpose to accomplish unsuperivsed dense prediction can be simplified with increasing $\mathbb{E}_{t \in T}[p(Y|t', m=q)p(t')]$. When $p(t')$ is assumed to be uniform distribution, it satisfies $\mathbb{E}_{t \in T}[p(Y|\text{do}(t))] \uparrow \propto \mathbb{E}_{t \in T}[p(Y|t', m=q)] \uparrow$ so that enhancing the likelihood of semantic groups $Y$ directly leads to increasing causal effect between $T$ and $Y$ even under the presence of $U$.

consists of independently learned patch feature points $y \in \mathbb{R}^r$. However, we cannot directly compute this likelihood as in standard supervised learning, primarily because there are no available pixel annotations. Instead, we substitute the likelihood of unsupervised dense prediction to concept-wise self-supervised learning based on Noise-Contrastive Estimation (Gutmann & Hyvärinen, 2010):

$$p(y \mid t', m) = \mathop{\mathbb{E}}_{y^+} \left[ \frac{\exp(\cos(y, y^+)/\tau)}{\exp(\cos(y, y^+)/\tau) + \sum_{y^-} \exp(\cos(y, y^-)/\tau)} \right], \qquad (4)$$

where $y, y^+, y^-$ denote anchor, positive, and negative features, and $\tau$ indicates temperature term.

**Positive & Negative Concept Selection.**   When selecting positive and negative concept features for the proposed self-supervised learning, we use a pre-computed distance matrix $\mathcal{D}_M$ that reflects concept-wise similarity between all $k$ concept prototypes such that $\mathcal{D}_M = \cos(M, M) \in \mathbb{R}^{k \times k}$ in concept clusterbook $M$. Specifically, for the given patch feature $t \in \mathbb{R}^c$ as an anchor, we can identify the most similar concept $q \in \mathbb{R}^c$ and its index: $\mathrm{id}_q$ such that $q = \arg\max_{m \in M} \cos(t, m)$. Subsequently, we use the anchor index $\mathrm{id}_q$ to access all concept-wise distances for $k$ concept prototypes within $M$ through $\mathcal{D}_M[\mathrm{id}_q, :] \in \mathbb{R}^k$ as pseudo-code-like manner. By using a selection criterion based on the distance $\mathcal{D}_M$, we can access concept indices for whole patch features to distinguish positive and negative concept features for the given anchor. That is, once we find patch feature points in $T$ satisfying $\mathcal{D}_M[\mathrm{id}_q, :] > \phi^+$ for the given anchor $t$, we designate them as positive concept feature $t^+$. Similarly, if they meet the condition $\mathcal{D}_M[\mathrm{id}_q, :] < \phi^-$, we categorize them as negative concept feature $t^-$. Here, $\phi^+$ and $\phi^-$ represent the hyper-parameters for positive and negative relaxation, which are both set to 0.3 and 0.1, respectively. Note that, we opt for soft relaxation when selecting positive concept features because the main purpose of our unsupervised setup is to group subdivided concept prototypes into the targeted broader-level categories. In this context, a soft positive bound is advantageous as it facilitates a smoother consolidation process. While, we set tight negative relaxation for selecting negative concept features, which aligns with findings in various studies (Khosla et al., 2020; Kalantidis et al., 2020; Robinson et al., 2021; Wang et al., 2021a) emphasizing that hard negative mining is crucial to advance self-supervised learning.

In the end, after choosing in-batch positive and negative concept features $t^+$ and $t^-$ for the given anchor $t$, we sample positive segmentation features $y^+$ and negative segmentation features $y^-$ from the concept-matched $Y = \{y \in \mathbb{R}^r\}^{hw}$ within the same spatial location as the selected concept features. Through the concept-wise self-supervised learning in Eq. (4), we can then guide the segmentation head $S$ to enhance the likelihood of semantic groups $Y$. We re-emphasize that for the given anchor feature (*head*), our goal of USS is the feature consolidation corresponding to positive concept features (*torso, hand, leg, etc.*), and the separation corresponding to negative concept features (*sky, water, board, etc.*), in order to achieve the targeted broader-level semantic groups (*person*).

**Concept Bank: Out-batch Accumulation.**   Unlike image-level self-supervised learning, unsupervised dense prediction requires more intricate pixel-wise comparisons, as discussed in Zhang et al. (2021). To facilitate this, we establish a concept bank, similar to He et al. (2020) but notably at a pixel-level scale, to accumulate out-batch concept features for additional comparison pairs. Following the same selection criterion as described above, we dynamically sample in-batch features in each training iteration and accumulate them into the concept bank $Y_{\text{bank}} \in \mathbb{R}^{k \times b \times r}$ for continuously utilizing other informative feature from out-batches, where $b$ represents the maximum number of feature points saved for each concept in $M \in \mathbb{R}^{k \times c}$. We incorporate these additional positive and negative concept features into the sets of $y^+$ and $y^-$ for the concept-wise self-supervised learning. Here, creating a concept bank can be seen as incorporating global views into the pixel-level self-supervised learning beyond local views, which also corresponds to considering all feature representations $T' \in \mathbb{R}^{n \times hw \times c}$ ($n$: total number of images in dataset) for frontdoor adjustment. As a concept bank update strategy, we implement random removal of 50% of the bank's patch features for each concept prototype, followed by random sampling of 50% new patch features into the concept bank at every training iteration. In addition, to perform stable self-supervised learning, we employ: (i) using log-probability not to converge to near-zero value due to numerous multiplication of probabilities: $\frac{1}{|Y|} \log p(Y|t', m) = \frac{1}{|Y|} \log \prod_{y \in Y} p(y|t', m) = \mathbb{E}_{y \in Y}[\log p(y|t', m)]$, and (ii) utilizing exponential moving average (EMA) on teacher-student structure, all of which have been widely used by recent self-supervised learning frameworks such as Grill et al. (2020); Chen et al. (2021); Caron et al. (2021); Zhou et al. (2022); Assran et al. (2022). Please see complete details of Step 2 procedure in Algorithm 2 and Appendix B.

Table 1: Comparing quantitative results and applicability to other self-supervised methods on CAUSE.

(a) Experimental results on COCO-Stuff.

| Method ($\mathbb{C} = 27$) | Backbone | mIoU | pAcc |
|---|---|---|---|
| IIC (Ji et al., 2019) | ResNet18 | 6.7 | 21.8 |
| PiCIE (Cho et al., 2021) | ResNet18 | 14.4 | 50.0 |
| SegDiscover (Huang et al., 2022) | ResNet50 | 14.3 | 40.1 |
| SlotCon (Wen et al., 2022) | ResNet50 | 18.3 | 42.4 |
| HSG (Ke et al., 2022) | ResNet50 | 23.8 | 57.6 |
| ReCo+ (Shin et al., 2022) | DeiT-B/8 | 32.6 | 54.1 |
| DINO (Caron et al., 2021) | ViT-S/16 | 8.0 | 22.0 |
| + STEGO (Hamilton et al., 2022) | ViT-S/16 | 23.7 | 52.5 |
| + HP (Seong et al., 2023) | ViT-S/16 | 24.3 | 54.5 |
| + **CAUSE-MLP** | ViT-S/16 | 25.9 | 66.3 |
| + **CAUSE-TR** | ViT-S/16 | **33.1** | **70.4** |
| DINO (Caron et al., 2021) | ViT-S/8 | 11.3 | 28.7 |
| + ACSeg (Li et al., 2023) | ViT-S/8 | 16.4 | - |
| + TranFGU (Yin et al., 2022) | ViT-S/8 | 17.5 | 52.7 |
| + STEGO (Hamilton et al., 2022) | ViT-S/8 | 24.5 | 48.3 |
| + HP (Seong et al., 2023) | ViT-S/8 | 24.6 | 57.2 |
| + **CAUSE-MLP** | ViT-S/8 | 27.9 | 66.8 |
| + **CAUSE-TR** | ViT-S/8 | **32.4** | **69.6** |
| DINO (Caron et al., 2021) | ViT-B/8 | 13.0 | 42.4 |
| + DINOSAUR (Seitzer et al., 2023) | ViT-B/8 | 24.0 | - |
| + STEGO (Hamilton et al., 2022) | ViT-B/8 | 28.2 | 56.9 |
| + **CAUSE-MLP** | ViT-B/8 | 34.3 | 72.8 |
| + **CAUSE-TR** | ViT-B/8 | **41.9** | **74.9** |

(b) Experimental results on Cityscapes.

| Method ($\mathbb{C} = 27$) | Backbone | mIoU | pAcc |
|---|---|---|---|
| IIC (Ji et al., 2019) | ResNet18 | 6.4 | 47.9 |
| PiCIE (Cho et al., 2021) | ResNet18 | 10.3 | 43.0 |
| SegSort (Hwang et al., 2019) | ResNet101 | 12.3 | 65.5 |
| SegDiscover (Huang et al., 2022) | ResNet50 | 24.6 | 81.9 |
| HSG (Ke et al., 2022) | ResNet50 | 32.5 | 86.0 |
| ReCo+ (Shin et al., 2022) | DeiT-B/8 | 24.2 | 83.7 |
| DINO (Caron et al., 2021) | ViT-S/8 | 10.9 | 34.5 |
| + TransFGU (Yin et al., 2022) | ViT-S/8 | 16.8 | 77.9 |
| + HP (Seong et al., 2023) | ViT-S/8 | 18.4 | 80.1 |
| + **CAUSE-MLP** | ViT-S/8 | 21.7 | 87.7 |
| + **CAUSE-TR** | ViT-S/8 | **24.6** | **89.4** |
| DINO (Caron et al., 2021) | ViT-B/8 | 15.2 | 52.6 |
| + STEGO (Hamilton et al., 2022) | ViT-B/8 | 21.0 | 73.2 |
| + HP (Seong et al., 2023) | ViT-B/8 | 18.4 | 79.5 |
| + **CAUSE-MLP** | ViT-B/8 | 25.7 | 90.3 |
| + **CAUSE-TR** | ViT-B/8 | **28.0** | **90.8** |

(d) Experimental results on Pascal VOC 2012.

| Method ($\mathbb{C} = 21$) | Backbone | mIoU |
|---|---|---|
| IIC (Ji et al., 2019) | ResNet18 | 9.8 |
| SegSort (Hwang et al., 2019) | ResNet101 | 11.7 |
| DenseCL (Wang et al., 2021b) | ResNet50 | 35.1 |
| HSG (Ke et al., 2022) | ResNet50 | 41.9 |
| MaskContrast (Van Gansbeke et al., 2021) | ResNet50 | 35.0 |
| MaskDistill (Van Gansbeke et al., 2022) | ResNet50 | 48.9 |
| DINO (Caron et al., 2021) | ViT-S/8 | - |
| +TransFGU (Yin et al., 2022) | ViT-S/8 | 37.2 |
| +ACSeg (Li et al., 2023) | ViT-S/8 | 47.1 |
| +**CAUSE-MLP** | ViT-S/8 | 46.0 |
| +**CAUSE-TR** | ViT-S/8 | **50.0** |
| DINO (Caron et al., 2021) | ViT-B/8 | - |
| +DeepSpectral (Melas-Kyriazi et al., 2022) | ViT-B/8 | 37.2 |
| +DINOSAUR (Seitzer et al., 2023) | ViT-B/8 | 37.2 |
| +Leopart (Ziegler & Asano, 2022) | ViT-B/8 | 41.7 |
| +COMUS (Zadaianchuk et al., 2023) | ViT-B/8 | 50.0 |
| +**CAUSE-MLP** | ViT-B/8 | 47.9 |
| +**CAUSE-TR** | ViT-B/8 | **53.3** |

(c) Self-supervised methods with CAUSE-TR.

| Dataset | Self-Supervised Methods | Backbone | mIoU | pAcc |
|---|---|---|---|---|
| COCO-Stuff | | | 45.3 | 78.0 |
| Cityscapes | DINOv2 (Oquab et al., 2023) | ViT-B/14 | 29.9 | 89.8 |
| Pascal VOC | | | 53.2 | 91.5 |
| COCO-Stuff | | | 39.5 | 73.8 |
| Cityscapes | iBOT (Zhou et al., 2022) | ViT-B/16 | 23.0 | 89.1 |
| Pascal VOC | | | 53.4 | 89.6 |
| COCO-Stuff | | | 34.1 | 72.1 |
| Cityscapes | MSN (Assran et al., 2022) | ViT-S/16 | 21.2 | 89.1 |
| Pascal VOC | | | 30.2 | 84.2 |
| COCO-Stuff | | | 21.5 | 59.1 |
| Cityscapes | MAE (He et al., 2022) | ViT-B/16 | 12.5 | 82.0 |
| Pascal VOC | | | 25.8 | 83.7 |

## 4 EXPERIMENTS

### 4.1 EXPERIMENTAL DETAILS

**Inference.** In inference phase for USS, STEGO (Hamilton et al., 2022) and HP (Seong et al., 2023) equally perform the following six steps: (a) learning $\mathbb{C}$ cluster centroids (Caron et al., 2018) from the trained segmentation head output where $\mathbb{C}$ denotes the number of categories in dataset, (b) upsampling segmentation head output to the image resolution, (c) finding the most closest centroid indices to the upsampled output, (d) refining the predicted indices through Fully-connected Conditional Random Field (CRF) (Krähenbühl & Koltun, 2011) with 10 steps, (e) Hungarian Matching (Kuhn, 1955) for alignment with CRF indices and true labels, and (f) evaluating mean of intersection over union (mIoU) and pixel accuracy (pAcc). We follow the equal six steps with $S_{\text{ema}}$ of CAUSE.

**Implementation.** Following recent works, we adopt DINO as an encoder baseline and freeze it, where the feature dimension $c$ of $T$ depends on the size of ViT: small ($c = 384$) or base ($c = 768$). For hyper-parameter in the clusterbook, the number of concept $k$ in $M$ is set to 2048 to encompass concept prototypes from pre-trained features as much as possible. During the self-supervised learning, the number of feature accumulation $b$ in concept bank is set to 100. In addition, output dimension $r$ of segmentation head is set to 90 based on the dimension analysis (Koenig et al., 2023). For the segmentation head, we use two variations: (i) **CAUSE-MLP** with simple MLP layers as in Hamilton et al. (2022) and (ii) **CAUSE-TR** with a single layer transformer. Please see details in Appendix B.

**Datasets.** We mainly benchmark CAUSE with three datasets: COCO-Stuff (Caesar et al., 2018), Cityscapes (Cordts et al., 2016), and Pascal VOC (Everingham et al., 2010). COCO-Stuff is a scene texture segmentation dataset as subset of MS-COCO 2017 (Lin et al., 2014) with full pixel annotations of common *Stuff* and *Thing* categories. Cityscapes is an urban street scene parsing dataset with annotations. Following Ji et al. (2019); Cho et al. (2021), we use the curated 27 mid-level categories from label hierarchy for COCO-Stuff and Cityscapes. As an object-centric USS, we follow Van Gansbeke et al. (2022) and report the results of total 21 classes for PASCAL VOC.

Table 2: Comparing linear probing performance. Table 3: Results of CAUSE with larger categories.

| | | COCO-Stuff | | Cityscapes | |
|---|---|---|---|---|---|
| Method | Baseline | mIoU | pAcc | mIoU | pAcc |
| DINO (Caron et al., 2021) | ViT-S/8 | 33.9 | 68.6 | 22.8 | 84.6 |
| +HP (Seong et al., 2023) | ViT-S/8 | 42.7 | 75.6 | 30.6 | 91.2 |
| +CAUSE-MLP | ViT-S/8 | 46.4 | 77.3 | 35.2 | 92.1 |
| +CAUSE-TR | ViT-S/8 | **47.2** | **78.8** | **37.2** | **93.5** |
| DINO (Caron et al., 2021) | ViT-B/8 | 29.4 | 66.8 | 23.0 | 84.2 |
| +STEGO (Hamilton et al., 2022) | ViT-B/8 | 41.0 | 76.1 | 26.8 | 90.3 |
| +CAUSE-MLP | ViT-B/8 | 48.3 | 79.8 | 38.2 | 93.4 |
| +CAUSE-TR | ViT-B/8 | **52.3** | **80.1** | **40.2** | **94.5** |

| | Method | Backbone | mIoU | pAcc |
|---|---|---|---|---|
| COCO-81 | MaskContrast (Van Gansbeke et al., 2021) | ResNet50 | 3.7 | 8.8 |
| | TransFGU (Yin et al., 2022) | ViT-S/8 | 12.7 | 64.3 |
| | **CAUSE-MLP** | ViT-S/8 | 19.1 | **78.8** |
| | **CAUSE-TR** | ViT-S/8 | **21.2** | 75.2 |
| COCO-171 | IIC (Ji et al., 2019) | ResNet50 | 2.2 | 15.7 |
| | PiCIE (Cho et al., 2021) | ResNet50 | 5.6 | 29.8 |
| | TransFGU (Yin et al., 2022) | ViT-S/8 | 12.0 | 34.3 |
| | **CAUSE-MLP** | ViT-S/8 | 10.6 | 44.9 |
| | **CAUSE-TR** | ViT-S/8 | **15.2** | **46.6** |

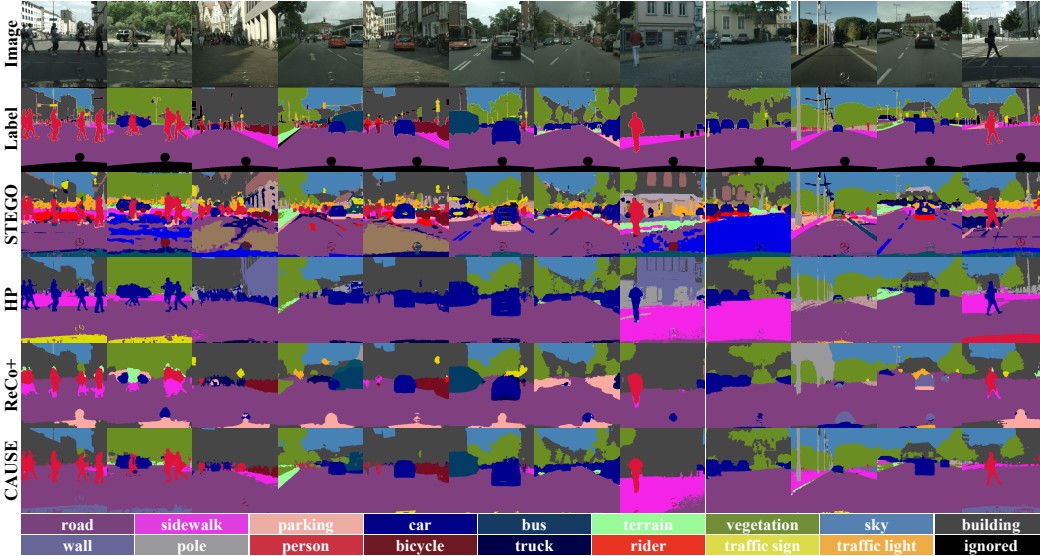

| road | sidewalk | parking | car | bus | terrain | vegetation | sky | building |
|---|---|---|---|---|---|---|---|---|
| wall | pole | person | bicycle | truck | rider | traffic sign | traffic light | ignored |

Figure 4: Qualitative comparison of unsupervised semantic segmentation for Cityscapes dataset.

## 4.2 VALIDATING CAUSE

**Quantitative & Qualitative Results.** We validate CAUSE by comparing with recent USS frameworks using mIoU and pAcc on various datasets. Table 1 (a) and (b) show CAUSE generally outperforms HSG (Ke et al., 2022), TransFGU (Yin et al., 2022), STEGO (Hamilton et al., 2022), HP (Seong et al., 2023), and ReCo+ (Shin et al., 2022), and our method achieves state-of-the-art results without any external information. Table 2 shows another superior quantitative results of CAUSE for linear probing than baselines, which indicates competitive dense representation quality learned in unsupervised manners. Furthermore, Fig. 1 and Fig. 4 illustrate CAUSE effectively assembles different level of granularity (*head, torso, hand, leg*, etc.), into one semantically-alike group (*person*). Please see additional qualitative results, analyses, and failure cases in Appendix C.

**Applicability to Object-centric Semantic Segmentation.** Preceding works, rooted in object-centric semantic segmentation models (Van Gansbeke et al., 2021; Yin et al., 2022; Zadaianchuk et al., 2023), initially generate pseudo-labels that differentiate between foreground (objects) and background. This process is typically accomplished by using Mask R-CNN (He et al., 2017) and DeepLabv3 (Chen et al., 2017), or saliency maps from DeepUSPS (Nguyen et al., 2019). In contrast, STEGO and HP abstains from relying on any external information beyond self-supervised knowledge. Therefore, they inherently lack the capability to segment an image into two broad categories: objects and a single background category, making them unsuitable for direct application to object-centric semantic segmentation. However, we highlight that simply adjusting smoother positive relaxation in CAUSE enables to discern background from foreground without any external information. The results of Pascal VOC 2012 is shown in Table 1(d), and its figures are illustrated in Appendix C.

**Generalization Capability** We first incorporate alternative self-supervised methods as our baseline, replacing DINO (Caron et al., 2021). In Table 1(c), we present an overview of adaptability in CAUSE across DINOv2 (Oquab et al., 2023), iBOT (Zhou et al., 2022), MSN (Assran et al., 2022), and MAE (He et al., 2022). Furthermore, we extend the number of clusters in CAUSE by utilizing MS-COCO 2017 (Lin et al., 2014), which comprises 80 object categories and one background category: (object-centric) COCO-81, and 171 categories encompassing both *Stuff* and *Thing* categories: COCO-171. Note that, positive $\phi^+$ relaxation is set to $0.4$ and $0.55$ respectively. Table 3 highlights CAUSE retains superior performances for USS even with larger categories. Especially, TransFGU (Yin et al.,

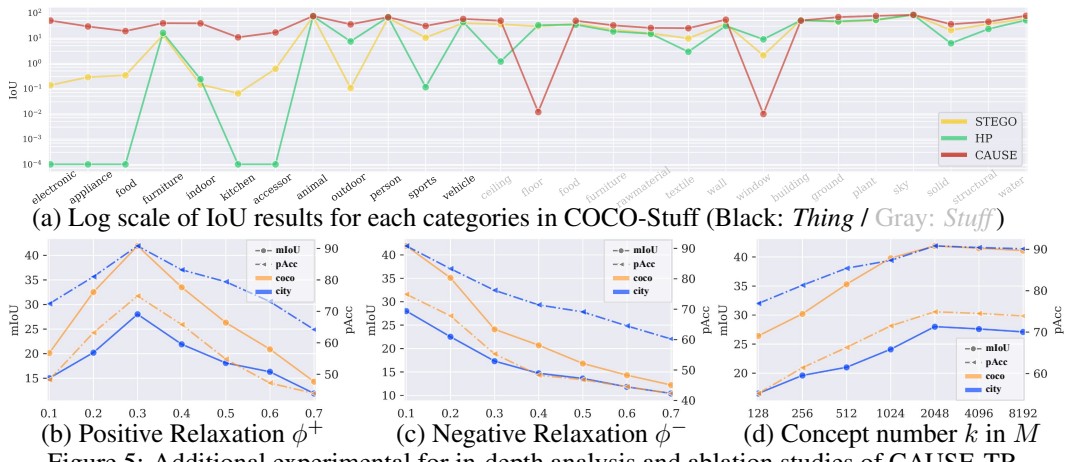

(a) Log scale of IoU results for each categories in COCO-Stuff (Black: *Thing* / Gray: *Stuff*)

(b) Positive Relaxation $\phi^+$ (c) Negative Relaxation $\phi^-$ (d) Concept number $k$ in $M$

Figure 5: Additional experimental for in-depth analysis and ablation studies of CAUSE-TR.

Table 4: Quantitative ablation results by controlling the other three factors of CAUSE-TR on ViT-B/8.

| (%) | | | CAUSE-MLP | | | | CAUSE-TR | | | |
| | | | COCO-Stuff | | Cityscapes | | COCO-Stuff | | Cityscapes | |
| Method of Concept Discretization | Bank | CRF | mIoU | pAcc | mIoU | pAcc | mIoU | pAcc | mIoU | pAcc |
|---|---|---|---|---|---|---|---|---|---|---|
| Maximizing Modularity (Newman, 2006) | ✗ | ✗ | 24.9 | 54.1 | 15.8 | 75.6 | 27.8 | 57.3 | 17.3 | 79.2 |
| | ✓ | ✗ | **31.3** | **69.0** | **25.3** | **89.5** | **39.5** | **72.5** | **28.8** | **90.7** |
| | ✗ | ✓ | 27.5 | 57.9 | 17.3 | 78.8 | 30.3 | 60.1 | 19.6 | 82.1 |
| | ✓ | ✓ | **34.3** | **72.8** | **25.7** | **90.3** | **41.9** | **74.9** | **28.0** | **90.8** |
| K-Means++ (Arthur & Vassilvitskii, 2007) | ✓ | ✓ | 27.8 | 64.7 | 18.9 | 81.3 | 33.7 | 62.7 | 20.4 | 83.2 |
| Spectral Clustering (Von Luxburg, 2007) | ✓ | ✓ | 30.7 | 65.1 | 20.8 | 83.5 | 35.9 | 66.7 | 22.8 | 84.1 |
| Agglomerative Clustering (Müllner, 2011) | ✓ | ✓ | 31.4 | 67.9 | 22.2 | 84.0 | 37.7 | 68.1 | 24.5 | 86.3 |
| Ward-Hierarchical Clustering (Murtagh & Legendre, 2014) | ✓ | ✓ | 31.8 | 67.5 | 22.9 | 84.7 | 37.5 | 68.2 | 24.7 | 87.0 |

2022) used Grad-CAM (Selvaraju et al., 2017) for generating category-specific pseudo-labels, thereby keeping consistent mIoU performance compared with COCO-81 and COCO-171. Nonetheless, CAUSE has a great advantage to pAcc especially in COCO-171 without any external information.

**Categorical Analysis.** To demonstrate that CAUSE can effectively address the targeted level of semantic grouping, we closely examine IoU results for each category. By validating the IoU results on a logarithmic scale in Fig. 5(a), we can observe that STEGO and HP struggle with segmenting *Thing* categories in COCO-Stuff, which demands fine-grained discrimination among concepts within complex scenes. In contrast, CAUSE consistently exhibits superior capability in segmenting concepts across most categories. These results are largely attributed to the causal design aspects, including the construction of the concept clusterbook and concept-wise self-supervised learning among concept prototypes. Beyond the quantitative results, it is important to highlight again that CAUSE exhibits significantly improved visual quality in achieving targeted level of semantic groupings than baselines as in Fig. 1 and Fig. 4. We include further discussions and limitations in Appendix D.

**Ablation Studies.** We conduct ablation studies on six factors of CAUSE to identify where the effectiveness comes from as in Fig. 5 and Table 4: (i) positive $\phi^+$ and (ii) negative relaxation $\phi^-$, (iii) the number of concepts $k$ in $M$, (iv) the effects of concept bank $Y_{bank}$ and (v) fully-connected CRF, and (vi) discretizing methods for concept clusterbook $M$. Through the empirical results, we first observe the appropriate relaxation parameter plays a crucial role in determining the quality of self-supervised learning. Furthermore, unlike semantic representation-level pre-training (Bao et al., 2022), we find that the number of discretized concepts saturates after reaching 2048 for clustering. We also highlight the effects of concept bank, CRF, and modularity maximization for effective USS.

## 5 CONCLUSION

In this work, we propose a novel framework called CAusal Unsupervised Semantic sEgmentation (CAUSE) to address the indeterminate clustering targets that exist in unsupervised semantic segmentation tasks. By employing frontdoor adjustment, we construct the concept clusterbook as a mediator and utilize the concept prototypes for semantic grouping through concept-wise self-supervised learning. Extensive experiments demonstrate the effectiveness of CAUSE, resulting in state-of-the-art performance in unsupervised semantic segmentation. Our findings bridge causal perspectives into the unsupervised prediction, and improve segmentation quality without any pixel annotations.

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

# A EXPANSION OF RELATED WORKS

**Unsupervised Semantic Segmentation.**    One of the key challenges in unsupervised dense prediction is the need to learn semantic representations for each pixel without the guidance of labeled data. In an early work for unsupervised semantic segmentation (USS), Ji et al. (2019) introduced the IIC framework, which aims to maximize mutual information between feature representations obtained from augmented views. Subsequently, several methods have advanced the quality of segmentation by introducing inductive bias through cross-image correspondences (Hwang et al., 2019; Cho et al., 2021; Wen et al., 2022) or by incorporating saliency information in an end-to-end manner (Van Gansbeke et al., 2021; Ke et al., 2022).

More recently, the discovery of semantic consistency in pre-trained self-supervised frameworks at the feature attention map (Caron et al., 2021) has led to prevalent approaches. Hamilton et al. (2022) introduced a method that leverages pre-trained knowledge and distills this information into the unsupervised segmentation task. Following this, various works (Wen et al., 2022; Yin et al., 2022; Ziegler & Asano, 2022) have employed self-supervised representations as pseudo segmentation labels (Zadaianchuk et al., 2023; Li et al., 2023) or as pre-encoded representations to incorporate additional prior knowledge (Van Gansbeke et al., 2021; Zadaianchuk et al., 2023) into the segmentation frameworks.

Our work aligns with Hamilton et al. (2022); Seong et al. (2023) in terms of enhancing segmentation features solely with the pre-trained representation. However, we emphasize the presence of indeterminate clustering targets inherent in unsupervised segmentation tasks. Our qualitative and quantitative results have demonstrated that the absence of evident clustering targets leads to poor segmentation outcomes in unsupervised settings. These negative effects have not been adequately addressed in previous works within the existing literature. Accordingly, for the first time, we interpret the unsupervised segmentation task within the context of causality, effectively bridging discretized representation learning and contrastive learning within this task.

**Causal Inference.**    In recent years, numerous studies (Wang et al., 2020a; Zhang et al., 2020b; Schölkopf et al., 2021; Lv et al., 2022) have applied causal inference techniques in DNNs to estimate the true causal effects between treatments and outcomes of interest. The fundamental approach to achieve causal identification involves blocking backdoor paths induced from confounders.

In several computer vision methods have employed various causal approaches such as backdoor adjustment establishing explicit confounders (Tang et al., 2020a; Zhang et al., 2020a; Yue et al., 2020; Liu et al., 2022), mediation analysis (Tang et al., 2020b; Niu et al., 2021), and generating counterfactual augmentations for randomized treatment assignments (Agarwal et al., 2020; Yue et al., 2021; Wang et al., 2022) and have been successfully applied at task-specific levels. More recently, various works (Kim et al., 2023b; Lee et al., 2023) have demonstrated that the causal approaches can be applied into the more specific computer vision areas with more sophisticated theories.

However, one of the challenges of applying causal inference to computer vision tasks is the explicit definition of confounding variables and the full control of them. Accordingly, we utilize frontdoor adjustment (Pearl, 1995) which can identify causal effects without the requirement of observed confounders, but relatively less explored in the context of computer vision tasks (Yang et al., 2021b;a). Inspired by recent developments in discrete representation learning (Van Den Oord et al., 2017; Esser et al., 2021), we proactively build a discretized concept representation and serve it as an informative mediator, allowing us to establish criteria for identifying positive and negative samples for a given query pixel representation. Consequently, this approach facilitates the integration of discretized representation and self-supervised learning into the process of unsupervised semantic segmentation.

# B DETAILED IMPLEMENTAION OF CAUSE

We present a detailed description of a concrete implementation for CAUSE, expanding upon the algorithms outlined in the method section and providing additional implementation details not covered in the experiment section. To validate identifiable and reproducible performance described in the experiment section, one can access the trained parameters of CAUSE-MLP and CAUSE-TR, as well as their visual quality, through the code document available in the supplementary material.

## B.1 Maximizing Modularity

When generating a mediator to design the concept cluster book, we need to compute a cluster assignment matrix $\mathcal{C} \in \mathbb{R}^{hw \times k}$ as described in Algorithm 1 in our manuscript. However, a computational issue arises when $k$ becomes large, such as the value of $2048$ selected for the main experiments, thus computing the measure of *Modularity* $\mathcal{H}$ becomes computationally expensive. To address this issue, we utilize the trace property of exchanging the inner multiplication terms, thereby reducing the computational burden, which can be written as follows:

$$\mathcal{H} = \frac{1}{2e}\mathrm{Tr}\left(\underbrace{\mathcal{C}(T,M)^{\mathrm{T}}\left[\mathcal{A} - \frac{dd^{\mathrm{T}}}{2e}\right]\mathcal{C}(T,M)}_{\mathbb{R}^{k \times k}}\right) = \frac{1}{2e}\mathrm{Tr}\left(\underbrace{\mathcal{C}(T,M)\mathcal{C}(T,M)^{\mathrm{T}}}_{\mathbb{R}^{hw \times hw}}\left[\mathcal{A} - \frac{dd^{\mathrm{T}}}{2e}\right]\right). \quad (5)$$

However, directly calculating the above formulation can lead to very small *Modularity* due to multiplying very small two values from $\mathcal{C}(T,M)\mathcal{C}(T,M)^{\mathrm{T}}$, rendering it ineffective optimization. To overcome this, we use the hyperbolic tangent and temperature term, which can be written as:

$$\mathcal{H} \approx \frac{1}{2e}\mathrm{Tr}\left(\tanh\left(\frac{\mathcal{C}(T,M)\mathcal{C}(T,M)^{\mathrm{T}}}{\tau}\right)\left[\mathcal{A} - \frac{dd^{\mathrm{T}}}{2e}\right]\right), \quad (6)$$

in order to scale up this value while it can maintain the possible range of the multiplication $\mathcal{C}(T,M)\mathcal{C}(T,M)^{\mathrm{T}}$ from the following range:

$$0 \leq \tanh\left(\frac{\mathcal{C}(T,M)\mathcal{C}(T,M)^{\mathrm{T}}}{\tau}\right) < 1, \quad (7)$$

such that it satisfies $0 \leq \mathcal{C}(T,M)\mathcal{C}(T,M)^{\mathrm{T}} \leq 1$ originated from the definition of the clustering assignment: $0 \leq \mathcal{C}(T,M) = \max(0, \cos(T,M)) \leq 1$.

## B.2 Transformer-based Segmentation Head

We use a single layer transformer decoder inspired by Vaswani et al. (2017); Carion et al. (2020) to build segmentation head with self-attention (SA), cross-attention (CA), and feed forward network (FFN) with its 2048 inner-dimension by default hyper-parameter (Vaswani et al., 2017), where a single head attention is used on its enough performance. To explain the detailed propagation procedure for CAUSE-TR, we first show vector-quantization mechanism for the pre-trained feature representation $T = \{t \in \mathbb{R}^c\}^{hw}$ by replacing each patch feature point $t$ with each of the most closest concept features in $M$ as follows:

$$Q = \{q \in \mathbb{R}^c \mid q = \arg\max_{m \in M}\cos(t,m)\}^{hw}. \quad (8)$$

Next, $Q$ is first propagated in SA, and $Q$ and $T$ are considered as query and key/value, respectively in CA, and the output of CA is propagated in FFN. Note that, learnable positional embedding is used in both query/key of SA and query of CA as Carion et al. (2020) have carried out. One different structure is to adopt additional two MLP layers in order to reduce the dimension from $c$ (ViT-S:384, ViT-B:768) to $r$ (90) for segmentation head output $Y$. This is because Koenig et al. (2023) empirically demonstrate that higher dimension $r$ over 100 brings in gradual collapse of clustering performance derived from the curse of dimensionality (Assent, 2012).

## B.3 Anchor Selection

In line 5 of Algorithm 2, we describe that we sample anchor patch feature point $y$ in $Y = \{y \in \mathbb{R}^r\}^{hw}$. In reality, it is extremely inefficient to select all number $hw$ of the patch feature points in $Y$ to perform anchor points in self-supervised learning, because of the limitation of the resource-constrained hardware. Therefore, we use a high-computation-reduced technique of stochastic sampling only 6.25% points ($\frac{1}{4}^2 \times 100(\%)$) among the number $hw$ points in $Y$, where we randomly select one feature point whenever a window having kernel size $4 \times 4$ and stride 4 is sliding along with $Y$.

### B.4 Positive & Negative Concept Selection

In line 6 of Algorithm 2, we either describe that we sample positive and negative concept features $y^+, y^-$ in the set of $Y_{\text{ema}}$ and $Y_{\text{bank}}$ for the given anchor patch feature point $y$: it expresses $y^+, y^- \sim \{Y_{\text{ema}}, Y_{\text{bank}} \mid y\}$. First, we find the patch feature point $t$ corresponding to $y$ and then search the most closest concept $q = \arg\max_{m \in M} \cos(t, m)$ and its index $\text{id}_q$. Next, we filter the positive $y^+$ and negative $y^-$ concept features satisfying each condition on $\mathcal{D}_M[\text{id}_q, :] > \phi^+$ and $\mathcal{D}_M[\text{id}_q, :] < \phi^-$. Then, we sample all of the positive and negative concept features in the set of $Y_{\text{ema}}$ and $Y_{\text{bank}}$. Note that, there are a few case that the row vector in $\mathcal{D}_M$ has a minimum value over zero, thus we technically set hard negative relaxation to $0.1$, instead of $0$.

### B.5 Concept Bank

In line 10 of Algorithm 2, the concept bank $Y_{\text{bank}}$ follows a specific rule: not all of the segmentation features $Y_{\text{ema}}$ are collected, but they are instead $50\%$ re-sampled based on the most closest concept indices individually, where the concept bank collects a maximum of $100$ features per concept prototype. Before re-sampling, $50\%$ of $Y_{\text{bank}}$ is randomly discarded. Considering that we have the number $2048$ of concept prototypes, the concept bank stores total number $100 \times 2048$ of the segmentation features. This ensures that the concept bank contains a comprehensive collection of treatment candidates $T'$, providing the diversity of selecting positive and negative concept features.

### B.6 Image Resolution and Augmentation

For COCO-Stuff and Cityscapes, we equally follow data-augmentation method of STEGO (Hamilton et al., 2022) and HP (Seong et al., 2023) which employ five-crop with crop ratio of $0.5$ in full image resolution and resizes the cropped images to $224 \times 224$ for CAUSE-MLP in training phase. For inference phase, images are resized to $320 \times 320$ along the minor axis followed by center crops of each validation image. For CAUSE-TR, $320 \times 320$ image resolution is used to train segmentation head of a single layer transformer, because the same number of queries and learnable positional embeddings is used in training and inference phase. For Pascal VOC 2012, COCO-81, and COCO-171, we follow data-augmentation method of TransFGU (Yin et al., 2022) which employs multiple-crop with multiple ratio. A significant different point is that STEGO, HP, and TransFGU employ additional data-augmentation techniques, including Horizontal Flip, Color-Jittering, Gray-scaling, and Gaussian-Blurring as geometric and photometric transforms, but CAUSE utilizes Horizontal Flip only.

## C Additional Experiments

Due to page limitations, we are unable to include a comprehensive set of visual results for unsupervised semantic segmentation on multiple datasets in the main manuscript. In this additional section, we provide various examples primarily from four datasets and show the comparison results with baseline methods.

### C.1 Additional Qualitative Results

To provide further evidence of unsupervised semantic segmentation results, we include additional qualitative visual results in Fig.6 and Fig.8 for COCO-Stuff and Cityscapes, respectively. The entire experimental setup remains consistent with the main manuscript, and we compare our proposed method with recent unsupervised semantic segmentation baselines (Hamilton et al., 2022; Seong et al., 2023; Shin et al., 2022) that also utilize pre-trained DINO (Caron et al., 2021) feature representations.

Additionally, we present qualitative results for object-centric semantic segmentation by providing visualizations for the PASCAL VOC, COCO-81 and COCO-171 in Fig. 9 and Fig. 7, respectively. All of these datasets include an additional background class. While the negative relaxation is set to the same value of $0.1$, we have adjusted the positive relaxation to $0.2$, $0.4$, and $0.55$ for PASCAL VOC, COCO-81, and COCO-171 datasets, respectively. This modification is primarily due to account for the coarsely merged background class, as it aids in distinguishing the intricate integration of the background concepts.

### C.2 FAILURE CASES

Unsupervised semantic segmentation is considered fundamentally challenging due to the combination of the absence of supervision and the need for dense pixel-level predictions. Even if we successfully achieve competitive unsupervised segmentation performance, there are some failure cases in which CAUSE may produce inadequate segmentation quality.

In Fig. 10, we present failed segmentation with other baselines. One of the observations is the existence of noisy segmentation outcomes, especially in complex scenes. A straightforward solution is to adjust the larger number $k$ when constructing the concept clusterbook $M$. However, we have observed a trade-off between handling complex scenes and dealing with relatively easier examples. To fundamentally address this issue in future directions, we expect to explore pre-processing techniques and incorporate multi-scale feature extraction methods, as demonstrated in previous works such as Kirillov et al. (2019); Kim et al. (2023a); Ranftl et al. (2021). These approaches aim to enhance the precision of detailed dense prediction.

Additionally, we have observed instances where the cluster assignments were incorrectly predicted for certain object instances. We believe that these failures can primarily be attributed to two factors: (i) the inherent limitations of leveraging pre-trained self-supervised frameworks originally designed for image-level prediction tasks, and (ii) the possibility of incompleteness in the employed clustering methods. As part of our future work, we believe it is essential to utilize foundation networks specifically designed for dense prediction tasks and clustering algorithms that can operate in an unsupervised manner. This approach will likely lead to more robust performance for unsupervised semantic segmentation.

### C.3 CONCEPT ANALYSIS IN CLUSTERBOOK

In the proposed causal approach in unsupervised semantic segmentation, we define discretized representation as a mediator (*concept clusterbook*) and leverage the advantages of discretization to facilitate concept-wise self-supervised learning through frontdoor adjustment. A natural question would be: *What is included in the clusterbook within the representation space?* To address the question, we conduct additional experiments that focused on retrieving concepts using the shared index of clusterbook. Firstly, we select an anchor index from the total 2048 concepts in clusterbook. Then, we retrieve image regions that corresponds to the same cluster index as the anchor. Furthermore, to merge more wider image regions considering pixel-level clusterbook indices, we employ the concept distance matrix as explained in Section 3.3. Specifically, we merge image regions based on their discretized index when the concept distance with the anchor index exceeds positive relaxation of softly 0.3. The retrieved results can be found in Fig. 11.

## D DISCUSSIONS AND LIMITATIONS

**Bootstrapping Pre-trained Models.** It is significantly challenging to handle fine-grained and complex scenes when dealing with unsupervised semantic segmentation using pre-trained feature representation. Based on the fact that the pre-trained features are designed to capture high-level semantic information, STEGO (Hamilton et al., 2022), HP (Seong et al., 2023), and ReCo+ (Shin et al., 2022) struggle yet fail to precisely segment intricate details within images, especially in scenarios with densely packed objects, complex backgrounds, or small objects as observed in Fig. 6-9. This is because the pre-trained models, originally designed for tasks like image classification or object detection, are not perfectly matched to understand the different level of granularity required for fine-grained segmentation. In contrast, our novel framework, CAUSE, bootstraps the knowledge of high-level pre-trained feature representation to achieve semantic grouping at pixel-level by bridging a causal approach combining the discrete concept representation with concept-wise self-supervised learning.

**Heuristically Static Hyper-Parameter.** CAUSE carefully assembles the concept clusterbook $M$, in advance, considering which concept features should be amalgamated or distinguished based on the intricate concept-wise distance matrix $\mathcal{D}_M$. One of nuisances involves heuristically establishing selection criterion for positive $\phi^+$ and negative $\phi^-$ relaxation, allowing for the construction of different level of granularity within the semantic groups $Y$. However, tailoring these criterion to

the specifics of a dataset can be a challenging endeavor. Grid-search, ranging ambitiously from $0.1$ to $0.7$, is employed in the quest to find optimal relaxation values, but such task demands heuristic efforts. Moreover, in the realm of image processing, adapting to dynamic environmental contexts within images, encompassing scenarios such as the presence of small objects or intricate scenes, is imperative. In future direction, it requires more dynamical process of selecting criterion, particularly for such specialized and complex contexts.

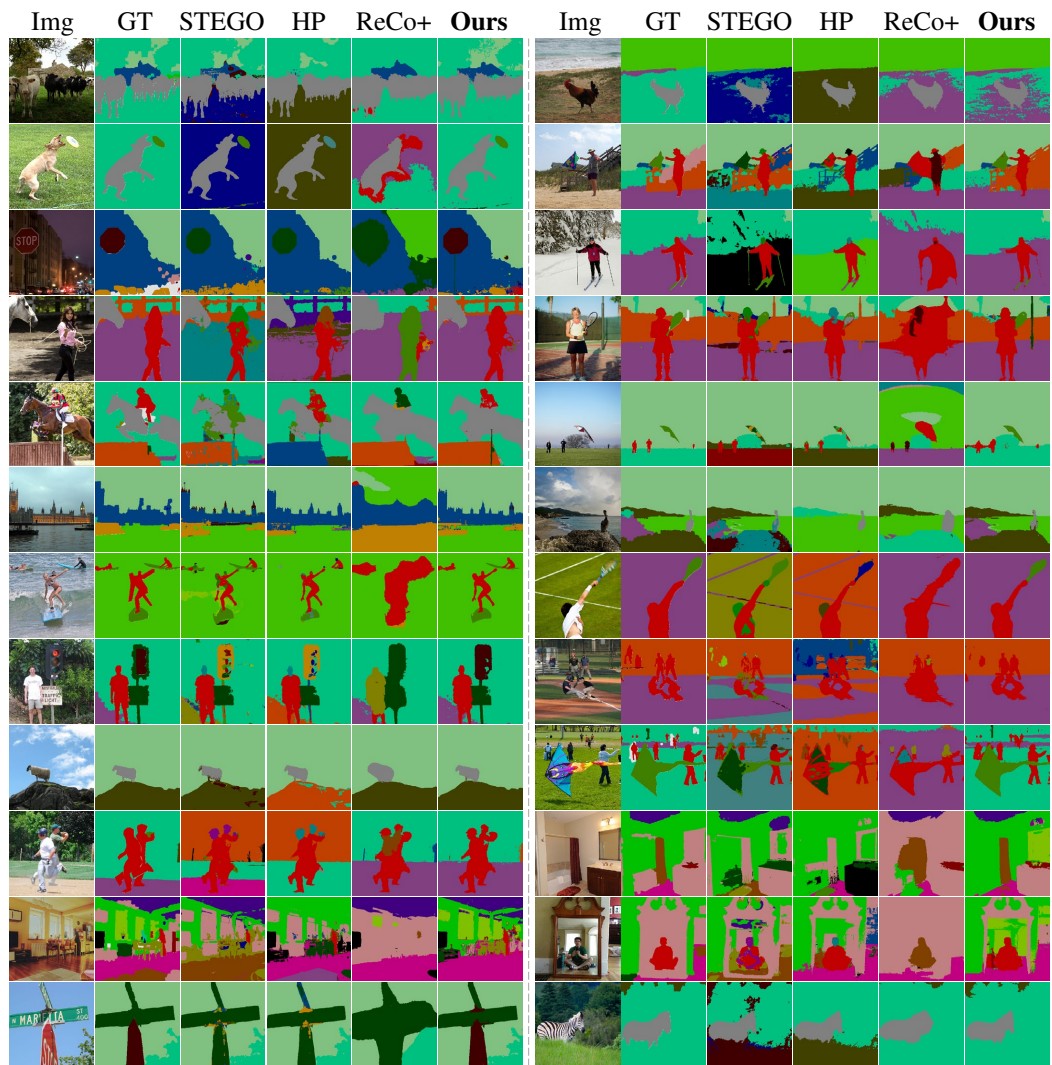

Figure 6: Additional qualitative results of unsupervised semantic segmentation for Coco-Stuff. Please look up the object color maps in the main manuscripts.

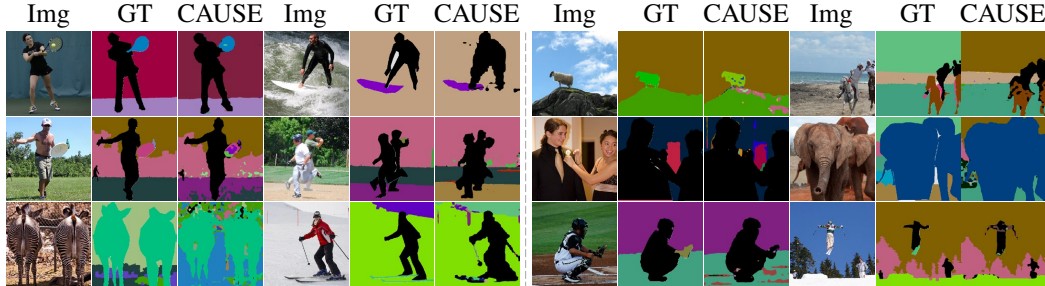

Figure 7: Qualitative results of unsupervised semantic segmentation for COCO-171, which is specialized for object-centric semantic segmentation with 171 categories.

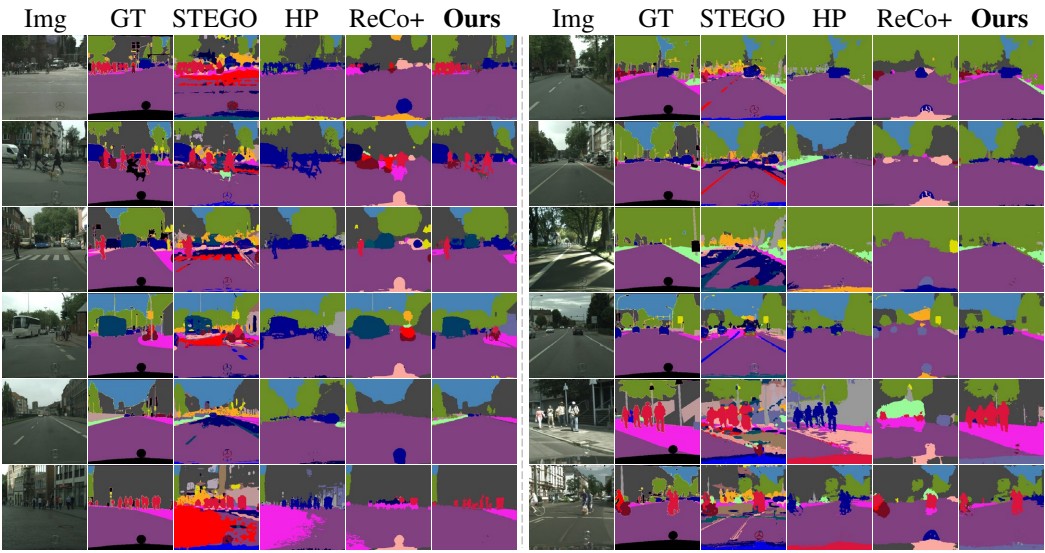

Figure 8: Additional qualitative results of unsupervised semantic segmentation for Cityscapes. Please look up the object color maps in the main manuscripts.

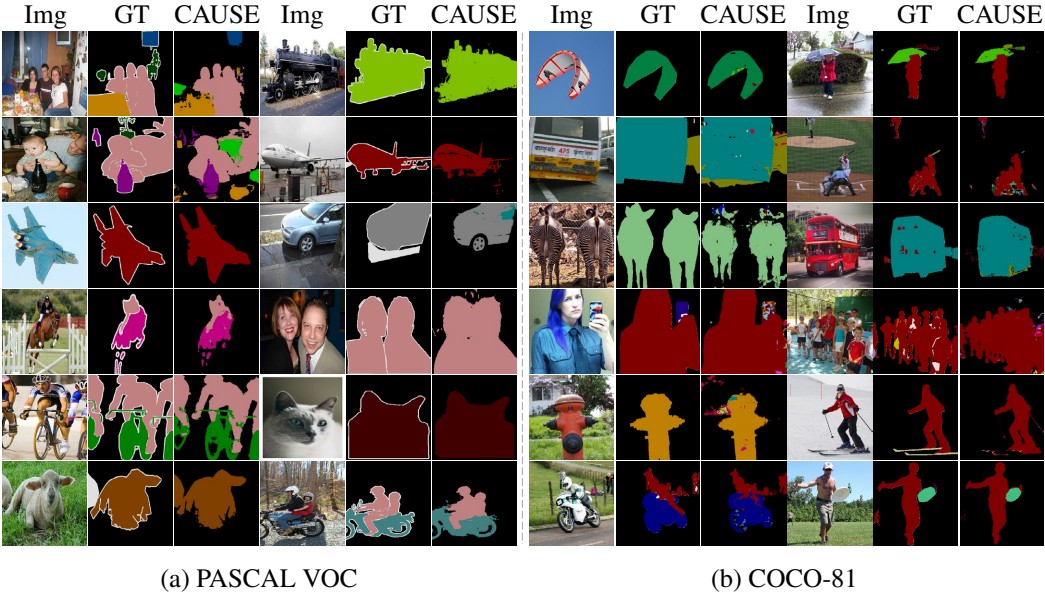

(a) PASCAL VOC            (b) COCO-81

Figure 9: Qualitative results of unsupervised semantic segmentation for PASCAL VOC and COCO-81, both of which are specialized for object-centric semantic segmentation.

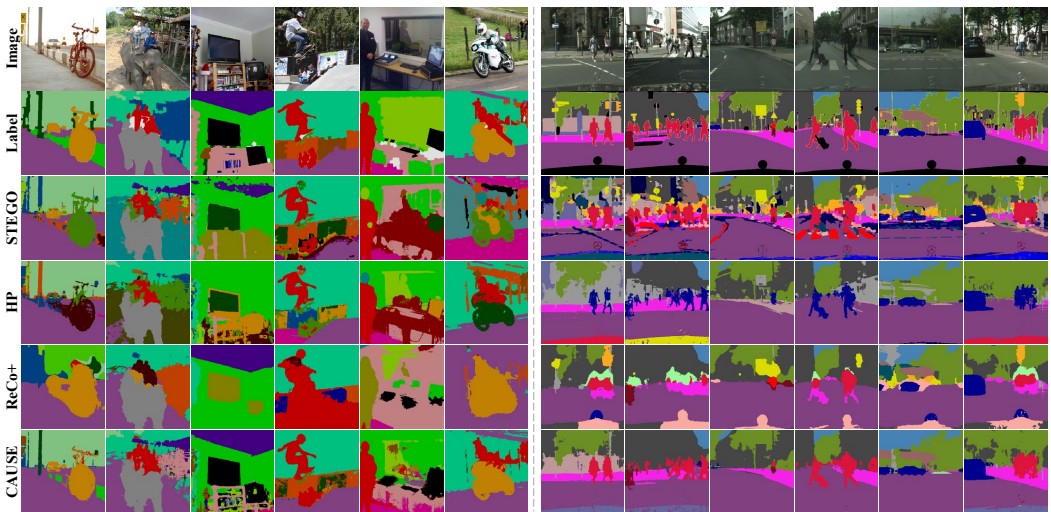

Figure 10: Failure cases of CAUSE and comparison results with other baselines.

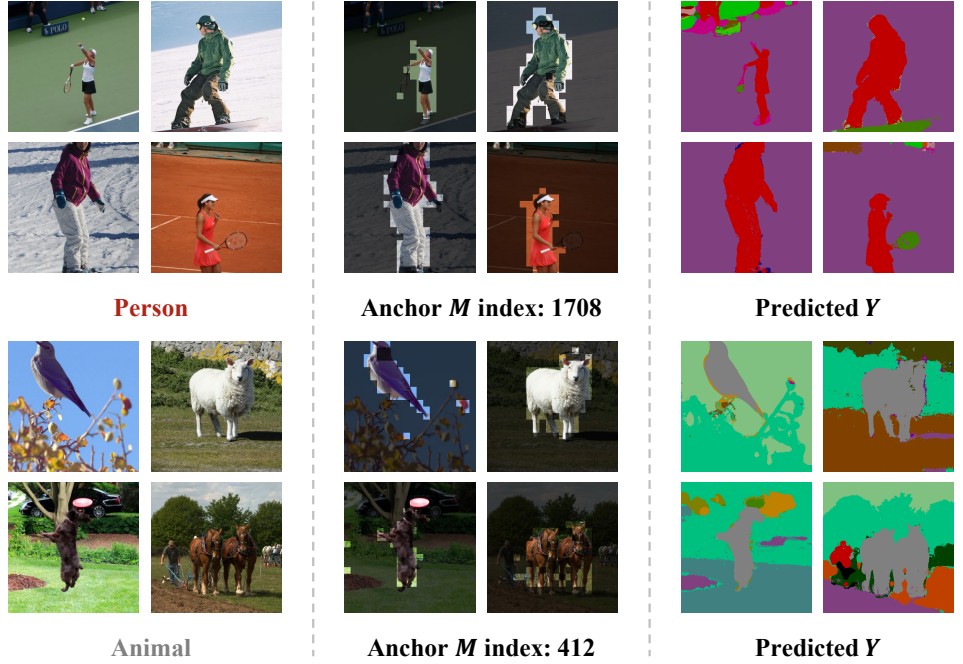

Figure 11: Retrieval results of the concept with respect to the shared index on clusterBook. We select *Person* and *Animal* categories and CAUSE prediction results on COCO-Stuff.

