# OpenReview forum: "Causal Unsupervised Semantic Segmentation"
_ICLR.cc/2024/Conference — Submitted to ICLR 2024_

### Official Review · Reviewer_vbSV · 2023-10-26

**Soundness:** 2 fair
**Presentation:** 2 fair
**Contribution:** 2 fair
**Rating:** 5
**Confidence:** 4

**Summary:**

This paper focuses on unsupervised semantic segmentation and proposes a framework based on causal inference. Specifically, the proposed framework employs two-step pipline to solve the task, which is claimed as intervention-oriented approach (i.e., frontdoor adjustment). The proposed method first constructs a concept clusterbook as a mediator and then adopts concept-wise self-supervised learning for pixel-level grouping. Extensive experiments are conducted on various datasets to demonstrate the proposed method.

**Strengths:**

- The proposed method achieves performance improvements on various datasets.
- The visualizations are rich and abundant.

**Weaknesses:**

- The causal diagram is not solid. Why the path $T \rightarrow Y$ could be omit?
- The specific explanation about $U$ is missing. It is not convincing to use question as definition.
- What are the specific representations of  $T,M,Y,U$? E.g., $T \in \mathbb{R}^{D\times H\times W}$.
- Is there any instance or example for explaining Figure 2?
- The authors claim that the main goal is to group semantic concepts that  meet the targeted level of granularity. How do the items in clusterbook correspond to various granularities?
- Is is feasible to direct evaulate the mIoU based on results of the concept with respect to the index on clusterBook?

**Questions:**

See Weaknesses*

---

> ### Author Response · Authors · 2023-11-15
>
> **Q1. The causal diagram is not solid. Why the path $T \rightarrow Y$ could be omit?**
>
> **A1.** We kindly argue that our causal diagram design is solid and it is highly related with unsupervised setup. We have not omitted $T \rightarrow Y$ pathway, this pathway means directly use the feature representation for semantic segmentation prediction, and it is equivalent with the vanilla DINO results in Table 1.
>
> The vanilla DINO results show that naively using $T$ is sub-optimal performance, thus we point out that the unawareness of the clustering targets in unsupervised setup makes difficulties during the segmentation.
>
> Accordingly, our design of causal diagram defines with an explicit ***“Mediator $M$"*** between $T$ and $Y$ ($T \rightarrow M \rightarrow Y$) to handle the difficulties in unsupervised learning. We highlight again that $M$ generates concept prototypes as priors with concrete causal background, and provides explicit distance criteria for the pixel grouping.
>
> ---
>
> **Q2. The specific explanation about $U$ is missing. It is not convincing to use question as definition.**
>
> **A2.** We respectfully highlight again that we have clearly stated $U$ as ***"How can we define what to cluster and how to do so under an unsupervised setting?"*** in Introduction. As we described, the $U$ indicates the lack of awareness regarding what and how to cluster for each pixel representation, especially when aiming for the desired level of granularity.
>
> For further understanding, let us assume that we have a pre-trained feature representation for unsupervised segmentation (from DINO). The main problem for conducting USS is that we don’t know ***“what concepts have to be grouped and segmented for semantic segmentation”***. As an extension of your Q1, just maximizing correspondence “continuous” feature representation ($T \rightarrow Y$) directly provides sub-optimal performance because grouping feature representation in continuous embedding space is challenging tasks to conduct, especially in the case we don’t know any segmentation information to cluster ($U$: what \& how to cluster). We hope this additional explanation can address any misunderstanding regarding $U$.
>
> ---
>
> **Q3. What are the specific representations of $T, M, Y, U$ ? e.g., $T\in \mathbb{R}^{D\times H \times W}$.**
>
> **A3.** Please remind that we have definitely specified each variables in detail though Section 3.2 to 3.3. Here, we specify again for the variables:
> $T\in\mathbb{R}^{hw \times c}$, $M\in\mathbb{R}^{k\times c}$, $Y=\lbrace y\in\mathbb{R}^{r}\rbrace^{hw}$. More importantly, we remind to the reviewer that $U$ is *"unobserved"* confounder.
>
> ---
>
> **Q4. Is there any instance or example for explaining Figure 2?**
>
> **A4.**  ***Yes, in Figure 1!*** The other baselines in the Figure 1 directly maximize “continuous” feature representation, thus suffering low segmentation quality with respect for the targeted-level objects in the images. Note that this is provoked due to $U$ in unsupervised setup.
>
> As stated in the caption of Figure 1, for instance, in column 4 of the figure, other baselines are suffering from segment human head and whole body part and often assigning different unsupervised predictions for these elements. Notably, our approach successfully group the targeted *person* objects, mainly due to the consolidation of feature representations from small fractions to the targeted level, utilizing concept prototypes in the clusterbook.
>
> ---
>
> **Q5. The authors claim that the main goal is to group semantic concepts that meet the targeted level of granularity. How do the items in clusterbook correspond to various granularities?**
>
> **A5.** The number of clusterbook is directly related with the number of concept prototypes in the images. During modularity maximization, $k$ indicates an explicit grouping number for the pixel representation, thus it is essentially related with granularity.
>
> As an extended Answer 4 of Reviewer zoBW, let us assume again there are a bunch of person images with same background. If we extremely set $k=2$, then the process of maximizing modularity may separate feature representation into only two distinct groups: *person* and *background*. If we gradually increase the number $k$, then the concept clusterbook will group the concept *person* into the more subdivide fractions such as *head, torso, hand, leg*, etc.
>
> The primary goal for clusterbook is on constructing as many subdivided concepts in advance (*Stage 1*), and providing consolidation criteria for the targeted granularity through adjusting the hyper-parameters of the positive ($\phi^{+}$) \& negative ($\phi^{-}$) relaxation (*Stage 2*). In addition, we would like to note that we have conducted comprehensive ablations for concept number $k$ in Figure 5(d) and concept retrieval analysis in Appendix C.3.
>
> ---

---

> ### Author Response · Authors · 2023-11-15
>
> **Q6. Is is feasible to direct evaulate the mIoU based on results of the concept with respect to the index on clusterBook?**
>
> **A6.** We would like to kindly remind that this is ***"unsupervised"*** evaluation. The evaluation protocol should utilize annotations in the datasets to directly measure the quality of unsupervised feature representation. That is the reason all other unsupervised baselines also use matching procedure (*e.g.,* Hungarian Matching) and probing prediction head with stop gradients ***during evalutations***. As you asked, it is possible to measure mIoU scores with respect to the clusterbook index. However, we cannot derive any meaningful evaluation outcomes for the cluterbook index, which is not ***TRUE*** annotations for the datasets.

---

> > ### Comment · Reviewer_vbSV · 2023-11-18
> > **Further Discussion**
> >
> > Thanks for the rebuttal of anthors.
> >
> > __Q1,Q2,Q3__: My major concern is still that the causal diagram is not solid.
> > On the one hand, the  U indicates "How can we define what to cluster and how to do so under an unsupervised setting".
> > On the other hand, the mediator M "generates concept prototypes as priors with concrete causal background, and provides explicit distance criteria for the pixel grouping".
> > But how can we define what to generate and how to do under an unsupervised setting for the mediator M?
> > Therefore, the mediator M also faces indeterminate as U.
> >
> > __Q4,Q5__. As clarified, "mainly due to the consolidation of feature representations from small fractions to the targeted level, utilizing concept prototypes in the clusterbook". But how to consolidate small fractions to the _targeted level_ in USS via clusterbook? Also, how to meet the _targeted level_ of granularity in USS? Is that finished by checking the evaluated performances?

---

> ### Author Response · Authors · 2023-11-18
>
> We appreciate to the reviewer for the reply and further discussion!
>
>
> ---
> We believe there may be a small misunderstanding for the role of $M$ and CAUSE consolidation process.
>
> We argue that construction $M$ is the way of handling $U$ in the pathway of $T \rightarrow Y$. We highlight again that, in Stage 1: Maximization Modularity, $M$ generates quantized prior knowledge from the pre-trained $T$ **as many as possible in advance ($k=2048$)**. $M$ is not related with the targeted granularity and solely optimized to represent $k$ number of concept prototypes from the pre-trained feature representation $T$. Therefore, ***construction of $M$ is not faced with the indeterminate $U$*** (what \& how to cluster). The primary goal for the quantized $M$ is on providing concept distance criteria for the subsequent Stage 2 contrastive learning. (*e.g., legs, head, torso, etc* in $M$ $\rightarrow$ the subsequent targeted *person* cluster).
>
> The process of consolidating subdivided concept fractions in $M$ to the targeted level is determined in Stage 2 (**NOT** in construction of $M$). In Stage 2, using contrastive learning, we control two hyper-parameter positive $\phi^{+}$ and negative $\phi^{-}$ relaxation based on the concept distances computed from $\mathcal{D}_{M}$ (please note that the number of targeted classes is defined by the number of classes in each dataset).
>
> For the evaluation protocol, mIoU is a measure of how well the assigned predictions are clustered, providing an indication of segmentation accuracy. On the other hand, pAcc is another metric that assesses the overall accuracy of the predicted pixel representations, offering a broader measure of pixel-wise correctness. We would like to highlight that the combined results of mIoU and pAcc for CAUSE demonstrate significant improvements compared to other baselines. It is important to note that if CAUSE were to fall short in achieving the targeted granularity, these improvements would not be attainable. The success of CAUSE is intricately tied to its ability to meet the specified granularity goals using our training strategy.
>
> Furthermore, we would like to kindly remind that we have included comprehensive analyses to validate that CAUSE exhibits strong generalization across various backbones and datasets and qualitative visualization for the various datasets (not only common datasets (cocostuff, cityscapes) but extended to the object-centric datasets (voc, coco)).
>
> We hope this further explanation can be helpful to understand the exact role of $M$ and how the concepts in $M$ is consolidated to the targeted-level we aim. Feel free to further discussion!
>
> ---

---

> > ### Comment · Reviewer_vbSV · 2023-11-19
> > **Further Discussion**
> >
> > In brief, how to ensure that M is independent with confounder U?
> > It is unclear, because the task of constructing prototypes is similar to USS, where the targets are indeterminate and we cannot generate all potential candidates.

---

> ### Author Response · Authors · 2023-11-19
>
> We thank to the reviewer for their active engagement!
>
> Firstly, we use self-supervised learning frameworks as backbone. The fundamental assumption behind using the pre-trained feature representation $T$ is on their property of semantic consistency for the object targets. Here, We would like to point out that when we optimize a clusterbook $M$, it is only related with $T$, that is $p(m|t)$ (***NOT*** considering of $Y$). We kindly remind that $U$ is only emerged during $T \rightarrow Y$ pathway!
>
> The role of $M$ is to provide potential concepts as many as possible using Modularity for the pre-trained feature representation $T$. The process of maximization of Modularity is one of common graph clustering methods such as k-NN, but more sophisticated approach for considering relations among vertices. We set larger number of $k=2048$ than the number of classes in targeted datasets (coco: 27 / cityscapes: 27). It cannot access any information of $Y$ during the modularity maximization, solely try to extract $2048$ representative quantized features from the pre-trained representation $T$. The objective is to construct a cluster book $M$ that ***captures essential concepts based on the intrinsic properties of $T$ without considering the specific segmentation targets $Y$***. This separation ensures that modularity maximization is dedicated to generating as many potential concept prototypes as possible that are semantically consistent with the features in $T$.
>
> Specifically, in Stage 1. (constructing $M$), we optimize the quantized embedding parameters of cluster book $M$ only using pre-trained representation $T$ for the specific datasets, which is equivalent with achieving maximizing modularity to generate concept prototypes. We train the $k$ prototypes for only one epoch.
>
> Please feel welcome any further questions. We believe that this discussion can enhance the clarity and quality of our contributions for potential readers.

---

> > ### Comment · Reviewer_vbSV · 2023-11-19
> > **Further Discussion**
> >
> > I have understood the implementation. But from the perspective of causal diagram, why is U only emerged in T->Y? Is it possible that U->M, considering that the targeted prototypes are indeterminate?

---

> ### Author Response · Authors · 2023-11-19
>
> No, it is out of scope our definition of $U$. The unobserved confounder $U$ is defined as indetermination during estimating **unsupervised prediction Y**. Thus $U$ cannot make any descendent causal path due to maximizing modularity only derived from $T$ (*i.e.,* $p(m|t)$). As we mentioned many times, constructing $M$ is ***NOT*** directly related with $Y$, thus not affected by $U$.
>
> More intuitively, $M$ is the way of handling $U$ for the case of we don't know what \& how to cluster to ***predict $Y$***. Constructing as many as possible implicit feature representation from $T$ does not involve any *what \& how to cluster* problem. Because the main purpose of $M$ is solely constructing $k$ concepts based on the pixel relations among vertices, ***NOT directly related with any granularity***.
>
> We hope this explanation can address our solid causal diagram design.

---

> ### Author Response · Authors · 2023-11-21
>
> We would like to add more detailed explanations for the reviewer's last question. We kindly highlight again that significant improvements of our proposed method on USS are specifically attributed to our robust causal design, leveraging Frontdoor adjustment.
>
> When, we conduct unsupervised-semantic-segmentation task in COCO-Stuff, we deal with 118000 number of training images in COCO-Stuff. It means we are going to cluster and separate features from $T\in\mathbb{R}^{118000\times h \times w \times c}$ with a highly striking dimension. In other words, if h=40 and w=40 in ViT-B/8, we must cluster and separate 118000$\times$ h $\times$ w = 188,800,000 number of features. Previous works: STEGO and HP have struggled to cluster very high-dimensional dimensional features $T$, therefore they somewhat have a seamless quality in easy classes (*e.g.,* sky, road), but they still fail to cluster harder classes (e.g., person, airplane) in complex scene. It is difficult to naively cluster and separate features from high-dimensional $T$ for all images. This is the reason why previous works have shown suboptimal quantitative and qualitative results, and fail to handle "what and how to cluster" $U$ for predicting $Y$.
>
> Instead, we introduce a guiding factor, namely Concept ClusterBool $M$ in Stage 1, for constructing subdiveded small concept fractions in advance to predict $Y$. We first quantize $T$ and then make Concept Clusterbook $M\in\mathbb{R}^{2048\times c}$ in stage 1 only using $T$. This 2048 number is significantly reduced number rather than 188,800,000 in $T$. Hence, in Stage 2, the factor $M$ can easily guide how the segmentation output feature $Y$ are clustered and separated, only with 2048 number of concepts and their concept-wise distance information (positive and negative relaxation, either). This is considered as a work of alleviating $U$ confounder’s affection to $Y$.
>
> In other words, $M$ plays a mediator role of helping to predict targeted-level of $Y$ in the subsequent Stage 2. For example, $Y$ in CAUSE-MLP is acquired from only $T$ such that $Y=S(T)$, but $Y$ is trained through $M$ (In step 2, it is related with $p(y|t’, m)$). That is, we would like to highlight that it is definitely different between “one helps to train $Y$” ($U\rightarrow M$ (X)) and “one is only a purpose of propagation and its output is $Y$” ($U\rightarrow M$ (O)).
>
> It is a guiding factor to make $Y$ be a successfully clustered features through its concepts, since the segmentation output $Y$ cannot be derived from $M$ directly!. Therefore, we would like to kindly remind that $M$ is independent with the confounder of “what and how to cluster”.

---

### Official Review · Reviewer_ivD5 · 2023-10-31

**Soundness:** 2 fair
**Presentation:** 2 fair
**Contribution:** 3 good
**Rating:** 6
**Confidence:** 4

**Summary:**

This paper discusses the task of unsupervised semantic segmentation (USS). The author proposes a new method called CAUSE, integrating USS into a causal problem through two steps: learning discrete sub-segmented representation with Modularity theory and conducting do-calculus with self-supervised learning in the absence of annotations. CAUSE bridges causal inference into unsupervised segmentation and obtain semantically clustered groups with the support of pre-trained feature representation. Extensive experiments on various datasets corroborate the effectiveness of CAUSE and achieve state-of-the-art results.

**Strengths:**

** The authors innovatively treat the USS task as a causal problem to solve the problem of determining the appropriate cluster level.

** The authors propose a discrete sub-segmented representation learning method using Modularity theory, which compensates for the lack of semantic understanding in traditional unsupervised segmentation methods.

** The authors introduce causal inference into the unsupervised segmentation task and enable semantic segmentation with self-supervised learning in the absence of annotations.

** The authors propose a concept drift detection method based on causal relationships, which can effectively detect the concept drift problem in unsupervised semantic segmentation.

**Weaknesses:**

** This paper lacks a detailed explanation of the specific methodologies used for constructing the concept clusterbook and conducting concept-wise self-supervised learning.

** The paper could benefit from a detailed explanation of the implementation details, such as the specific architectures used for the segmentation head and the pre-trained model.

** This paper does not discuss the limitations or potential drawbacks of the proposed framework, which would have been useful for readers to understand the scope and applicability of the approach.

** The comparison with recent and state-of-the-art methods in unsupervised semantic segmentation is missing, which could provide a comprehensive evaluation of the proposed framework.

**Questions:**

**  Please discuss the limitations or potential drawbacks of the proposed framework? It would be helpful for readers to understand the scope and applicability of the approach.

** Please provide more details on the implementation, such as the specific architectures used for the segmentation head and the pre-trained model?

**  Please discuss the computational complexity of the proposed framework? Please provide any insights or analysis on the computational efficiency of the approach?

---

> ### Author Response · Authors · 2023-11-15
>
> **Q1. This paper lacks a detailed explanation of the specific methodologies used for constructing the concept clusterbook and conducting concept-wise self-supervised learning. The paper could benefit from a detailed explanation of the implementation details, such as the specific architectures used for the segmentation head and the pre-trained model.**
>
> **A1.** Based on Answer 1 and 2 of Reviewer 1Vow and Answer 1 and 3 of Reviewer zoBW, we will add technical details and insightful writings included in Appendix, once we take a potential next step (cam-ready) due to extra-one page for enough space.
>
> However, in Appendix B, we have already described a concrete implementation for training details of CAUSE about (B.1) Maximizing Modularity, (B.2) Transformer-based Segmentation Head, (B.3) Anchor Selection, (B.4) Positive \& Negative Concept Selection, (B.5) Concept Bank, and (B.6) Image Resolution and Augmentation.
>
> Especially, Appendix B.2 ***"Transformer-based Segmentation Head"*** has included the propagation detail of this segmentation head and which paper inspired to this segmentation head. Moreover, we did not use pre-trained model for segmentation head but train from scratch. We would like to kindly remind the paragraph in Appendix B.2.
>
> ---
>
> **Q2. This paper does not discuss the limitations or potential drawbacks of the proposed framework, which would have been useful for readers to understand the scope and applicability of the approach.**
>
> **A2.** We would like to respectfully argue that we have described Appendix D ***"Discussion and Limitations"***. We kindly remind that as we stated in the last paragraph of *Categorical Analysis* of Sec 4.2: "We include further discussions and limitations in Appendix D".
>
> ---
>
> **Q3. The comparison with recent and state-of-the-art methods in unsupervised semantic segmentation is missing, which could provide a comprehensive evaluation of the proposed framework.**
>
> **A3.** We respectfully disagree that we have already compared CAUSE with SOTA methods for unsupervised semantic segmentation on Papers-With-Code. To the best of our knowledge, we have comprehensively compared with whole existing unsupervised semantic segmentation baselines: MaskContrast (*ICCV 2022*), HSG (*CVPR 2022*), ReCo+ (*NeurIPS 2022*), STEGO (*ICLR 2022*), TranFGU (*ECCV 2022*), DINOSAUR (*ICLR 2023*), HP (*CVPR 2023*). Please note that compared with STEGO and HP, we even extend the comparison targets to object-centric unsupervised semantic segmentation in Pascal VOC 2012 and COCO-81 and compare CAUSE performance with DeepSpectral (*CVPR 2022*), Leopart (*CVPR 2022*), ACseg (*CVPR 2023*), COMUS (*ICLR 2023*).
>
> Please reply specific missing references for the further comparison. We are willing to add them in the experiment tables in the main manuscript.
>
> ---

---

> ### Author Response · Authors · 2023-11-15
>
> **Q4. Please discuss the computational complexity of the proposed framework? Please provide any insights or analysis on the computational efficiency of the approach?**
>
> **A4.** Borrowing Answer 2 for the Question 2 Reviewer zoBW: (***Calculation of affinity matrix is time consuming?***), we will discuss the computational complexity of two step procedures.
>
> For Step 1, the potential computational burden can be thought of as calculation of affinity matrix over spatial resolution size in features. However, we calculate affinity matrix $\mathcal{A}$ in batch-wise manners. For example, we have pre-trained features $T\in\mathbb{R}^{b\times hw \times c}$ with batch number $b$. Then, its affinity matrix is nothing but $\mathcal{A}\in\mathbb{R}^{b \times hw \times hw}$. Even, we optimize the maximization of modularity through just ***"one epoch"*** with Adam optimizer, as we mentioned in the last paragraph of Sec 3.2. Furthermore, Appendix B.1, "Maximizing Modularity" has already described how to efficiently compute cluster assignment matrix using trace property. For our experiment environment (CPU: Intel(R) Xeon(R) Gold 6230R, GPU: RTX 3090 $\times$ 4EA, and RAM: 256GB), in the end, we take about within 10 min, 30 min, 45 min, 1 hour, and 10 min to perform modularity maximization on Cityscapes, COCO-Stuff, COCO-81, COCO-171, and Pascal-VOC 2012. Based on these computation time in modularity maximization, therefore, calculating affinity matrix and conducing modularity maximization requires negligible computation.
>
> For Step 2, the computationally expensive part is line 7 in Algorithm 2, which calculates the likelihood of semantic groups $Y$ (*i.e.,* segmentation head features) to act concept-wise self-supervised learning. This is because, once we select all of anchor feature points $y$ in $Y =\lbrace y\in\mathbb{R}^{r}\rbrace^{hw}$, we should collect all positive and negative features corresponding to each anchor feature point of which the number is $hw$. For example, we have segmentation head features $Y\in\mathbb{R}^{40(h)\times 40(w) \times 90(r)}$ and assume that no matter which anchor feature point $y$ in $Y$ is selected, there exist the $100$ number of positive ($p$) or negative ($n$) feature points, respectively. Then, we must deal with the number $40(h)\times 40(w) \times \lbrace100 (p)+100 (n)\rbrace = 320,000$ of the combinations between anchor feature points, positive, and negative features. Furthermore, if we consider batch-wise computation with $16$ batches, the highly striking number $320,000\times 16 = 5,120,000$ of combinations are needed. To effectively avoid this desperate computation, we *stochastically sampling* only $6.25$% of all anchor feature points as already illustrated in Appendix B.3 ***"Anchor Selection"***. This efficiently reduce $5,120,000 \times \frac{6.25}{100}=320,000$ which is surprisingly equal to one batch computation despite using full $16$ batches. From our effort trying to much lesson surplus computation, we can reach 30-45 min, 2-3 hour, 3-4 hour, 4-5 hour, and 30-45 min for Cityscapes, COCO-Stuff, COCO-81, COCO-171, and Pascal VOC 2012 with total two or three epochs. Note that, Distributed Data Parallel (DDP) [R1] with $16$ batches per GPU and Mixed Precision (float32-float16) [R2] are together used to technically more reduce computation in this environment setting: CPU: Intel(R) Xeon(R) Gold 6230R, GPU: RTX 3090 $\times$ 4EA, and RAM: 256GB.
>
> Beyond, the paragraph of *Implementation* in Sec 4.1 and Appendix B.2 ***"Transformer-based segmentation head"*** describe we use a single layer transformer-based segmentation head even with a single attention head. This is attributed to our primary intention making a simple yet effective model design for CAUSE. Furthermore, please see the attached code documents for full details and requirements.
>
> ---
>
> **References**
>
> * [R1] Pytorch distributed: Experiences on accelerating data parallel training. arXiv preprint arXiv:2006.15704 (2020).
>
> * [R2] Mixed Precision Training. International Conference on Learning Representations. 2018.

---

> > ### Author Response · Authors · 2023-11-21
> > **Remind Comment**
> >
> > We kindly remind the reviewer that the discussion period will conclude in the next 24 hours.
> >
> > We have comprehensivly answered to the raised questions, and we believe that many of them can be easily addressed with further explanation, some of which are already included in the Appendix.
> >
> > We are looking forward to further discussions and any additional feedbacks.

---

### Official Review · Reviewer_zoBW · 2023-10-31

**Soundness:** 3 good
**Presentation:** 3 good
**Contribution:** 2 fair
**Rating:** 5
**Confidence:** 4

**Summary:**

This paper proposed an unsupervised semantic segmentation method based on theory of causal inference. It introduce a concept clusterbook to serve as mediator to decide the cluster granularity. The cluster gradularity is considered a challenge and the key that effects unsupervised semantic segmentation.

**Strengths:**

1. overall the paper is well written.
2. The proposed pipeline of modularity clustering works well on benchmarks.

**Weaknesses:**

The link between proposed pipeline and causal inference is not clear, even though the authors  pays a lot of attention in explaining what's backdoor and frontdoor adjustments. In addition, some key details of the pipeline are not very clear, which requires further explainations.

**Questions:**

1. The link between the proposed method and the causal inference is not very clear. The authors do pay much attention on theory and formulation of frontdoor adjustment, however is there explicit link between it with the proposed pipeline? that is between equation 1 and the algorithm 1 and 2.

2. Calculation of affinity matrix is time consuming? how many samples does it use while calculating the affinity matrix? Does the codebook update while training? or fixed based on pretrained DINO features? In addition, how to guarantee the the codebook spans different levels, how to decide the propoer granularity?

3. STEGO has similar mechanism of contrastive learning. which module does the proposed method benefits more from?  The codebook or the ST learning?

4. A minor issue is that the ALGORITHM 1 should be placed near sec. 3.2 to prevent confusing

---

> ### Author Response · Authors · 2023-11-14
>
> **Q1. The link between the proposed method and the causal inference is not very clear. The authors do pay much attention on theory and formulation of frontdoor adjustment, however is there explicit link between it with the proposed pipeline? that is between equation 1 and the algorithm 1 and 2.**
>
> **A1.** We believe that the link between unsupervised semantic segmentation (USS) and causal inference is clearly explained in ***Causal Perspective on USS*** (Section 3.1) beyond the theoretical background. In Eq. (1), we would like to kindly remind that each underbraced step, denoted as Step 1 and Step 2, is directly correlated with Algorithm 1 and Algorithm 2, respectively.
>
> That is, Step 1 of Eq. (1) corresponds to $p(m|t)$ in Eq. (2), and Step 2 of Eq. (1) corresponds to $p(Y|t',m)p(t')$ in Eq. (2). Increasing both probabilities gets the probability $p(Y| \text{do}(T))$ of frontdoor adjustment to be improved.
>
> Here, Algorithm 1 is the procedure of finding concept prototypes $M$ in rich pre-trained features $T$, thus it is the efforts of improving $p(m|t)$ for all concepts $m \in M$ and all spatial feature points $t\in T$. Furthermore, Algorithm 2 is the procedure of enhancing the likelihood $p(Y|t', m)=\prod_{y\in Y}p(y| t', m)$, for which Eq. (4) represents $p(y| t', m)$ equals to the probability of Noise-Contrastive Estimation.
>
> Furtheremore, the footprint writing in page number 5 explains the concrete reasoning why maximizing the likelihood for the given $M$ is directly leading to increasing causal effect between $T$ and $Y$.
>
> ---
>
> **Q2. Calculation of affinity matrix is time consuming? how many samples does it use while calculating the affinity matrix?**
>
> **A2.** No. We calculate affinity matrix $\mathcal{A}$ in batch-wise manners. For example, we have pre-trained features $T\in\mathbb{R}^{b\times hw \times c}$ with batch number $b$. Then, its affinity matrix is nothing but $\mathcal{A}\in\mathbb{R}^{b \times hw \times hw}$. Even, we optimize the maximization of modularity through just ***one epoch*** with Adam optimizer, as we mentioned in the last paragraph of Sec 3.2. Furthermore, Appendix (B.1) ***"Maximizing Modularity"*** has already described how to efficiently compute modularity using trace property. For our experiment environment (CPU: Intel(R) Xeon(R) Gold 6230R, GPU: RTX 3090 $\times$ 4EA, and RAM: 256GB), in the end, we take about within 10 min, 30 min, 45 min, 1 hour, and 10 min to perform modularity maximization on Cityscapes, COCO-Stuff, COCO-81, COCO-171, and Pascal-VOC 2012. Based on these computation time in modularity maximization, therefore, calculating affinity matrix and modularity requires negligible computation.
>
> ---
>
> **Q3. Does the codebook update while training? or fixed based on pretrained DINO features?**
>
> **A3.** Concept clusterbook $M$ is trained only in Step 1 and fixed during Step 2. We would like to clarify its explicit mention in Sec 3.3.
>
> ---
>
> **Q4. How to guarantee the the codebook spans different levels, how to decide the propoer granularity?**
>
> **A4.** For the modularity maximization of pixel representation from DINO, the number of codebook ($k$) necessarily indicates the granularity level. For instance, let us assume that we have bunch of image samples of *person* with same background. In the extreme case ($k=2$), the codebook may include only two concepts, entire *person* and *background* concepts. On the contrary, with a larger $k$, the codebook would subdivide the *person* concept into subdivided components such as *head*, *torso*, *hands*, *legs*, and so forth. Here, determining the proper level of granularity is the most challenge in the context of ***"unsupervised"*** prediction. This is because somebody may want to group head and torso into upper body to divide person into upper body and lower body. Nobody knows the proper granularity until a criterion to grouping clusters is given (*i.e.,* label annotations).
>
> Here, our insight to handle the unsupervised setup is on collecting a large number $k$ of implicit concepts in advance using an explicit mediator (clusterbook) to span a broad spectrum of concept prototypes (*i.e.,* granularity) in advance during Step 1, and we consolidate and separate the concepts into the coarse-level semantics by *controlling only two hyper-parameter positive $\phi^+$ and negative $\phi^-$ relaxation* based on the concept distances $\mathcal{D}_M$ during Step 2. In other words, if COCO-Stuff's ground truth is changed by someone who wants to adjust another granularity criteria to cluster, then we need to control the two hyper-parameter to fit the semantic level someone desires. We further remind that we have conducted concept retrieval analysis for the clusterbook in Appendix C.3 and ablation study for the number of $k$ in Figure 5.
>
> ---

---

> ### Author Response · Authors · 2023-11-14
>
> **Q5. STEGO has similar mechanism of contrastive learning. Which module does the proposed method benefits more from? The codebook or the ST learning?**
>
> **A5.** ***"Codebook!"*** We highlight again that our improvements are mainly on the explicit mediator design, "Concept ClusterBook". The definite evidence is empirical validation for the ablation study of the concept number $k$ in Fig. 5(d). If Concept ClusterBook $M$ does not highly impact to unsupervised semantic segmentation (USS) performance, then the dramatic degraded performance cannot happen with fewer number of concepts in Fig. 5(d). In other words, Concept Clusterbook is so important that USS performance highly depends on the number of concepts.
>
> Moreover, the main purpose of student-teacher structure is to acquire stably high performance despite with small batch because its ensemble capacity [R1] on Polyak-Ruppert averaging with an exponential decay [R2, R3] bootstraps the model performance effectively [R4]. That is, it is more effective when we cannot enlarge batch size due to task-specific dataloader burden or hardware resource limitation but should prepare a collection of various queue like [R5]. From this reason, we introduce a concept bank possibly including numerous concepts of out-batch features, instead of enlarging number of batch size (hardware constraint). In brief, Concept ClusterBook is highly dependent on high USS performance, and student-teacher learning strategy with Concept Bank by exponential averaging helps to bootstrap stable USS performance despite with few batches.
>
> ---
>
> **Q6. A minor issue is that the ALGORITHM 1 should be placed near sec. 3.2 to prevent confusing**
>
> **A6.** We appreciate your great comments to make our paper further understandable to readers. We will definitely modify the location of Algorithm 1 to make it near to Sec 3.2, and we will add technical details and insightful writings in Sec 3 based on Answer 1 and 3, once we take a potential next step (cam-ready) due to extra-one page for enough space. Further, we will add more discussion content about Answer 2, 4, and 5 in Sec 4.
>
> ---
>
> **References**
>
> * [R1] Mean teachers are better role models: Weight-averaged consistency targets improve semi-supervised deep learning results. Advances in neural information processing systems 30 (2017).
>
> * [R2] Acceleration of stochastic approximation by averaging. SIAM journal on control and optimization 30.4 (1992): 838-855.
>
> * [R3] Ruppert, David. Efficient estimations from a slowly convergent Robbins-Monro process. Cornell University Operations Research and Industrial Engineering, 1988.
>
> * [R4] Bootstrap your own latent-a new approach to self-supervised learning. Advances in neural information processing systems 33 (2020): 21271-21284.
>
> * [R5] Momentum contrast for unsupervised visual representation learning. Proceedings of the IEEE/CVF conference on computer vision and pattern recognition. 2020.

---

> > ### Author Response · Authors · 2023-11-21
> > **Remind Comment**
> >
> > We kindly remind the reviewer that the discussion period will conclude in the next 24 hours.
> >
> > We have comprehensivly answered to the raised questions, and we believe that many of them can be easily addressed with further explanation, some of which are already included in the Appendix.
> >
> > We are looking forward to further discussions and any additional feedbacks.

---

### Official Review · Reviewer_1Vow · 2023-10-31

**Soundness:** 3 good
**Presentation:** 2 fair
**Contribution:** 3 good
**Rating:** 6
**Confidence:** 3

**Summary:**

This article introduces an algorithm CAUSE to address unsupervised semantic segmentation. The algorithm interprets the problem from a causal inference perspective, and mainly consists of two steps, i.e., building concept prototypes by maximising modularity and semantic grouping via concept-wise self-supervised learning. The algorithm has been evaluated on a set of segmentation datasets, like COCO-Stuff, Cityscapes, and VOC.

**Strengths:**

The causal inference perspective to solve unsupervised semantic segmentation is interesting and novel to me. The algorithm design basically makes sense, and shows promising performance on standard datasets.

**Weaknesses:**

My major concern revolves around Sec. 3.3. Writing in this part is not good. It is hard to understand all implementation details. __First__, it is unclear how Eq. 4 is derived. Since this is probably the most important part of the algorithm, more explanations should be given. __Second__, I am confused about how "find patch feature points in $T$ satisfying $\mathcal{D}_M[id_q,:]>\phi^+$" is implemented in practice. As far as I understand, $\mathcal{D}_M$ only summarizes prototype-prototype similarities; then how to select features based on the aforementioned rule? __Third__, for the statement "we set tight negative relaxation ..., ... emphasizing that hard negative mining is crucial to advance self-supervised learning", does a tight negative relaxation indicate easier negative mining?

The concept prototype bears high similarity with the work [ref1]. I am wondering whether the sinkhorn-knopp algorithm used in [ref1] can be used here for prototype generation.

From Fig. 4, it appears that the method works particularly well in object boundaries (very close to ground-truths for, e.g., persons). While this is impressive for unsupervised segmentation, I am curious how the algorithm improves over other methods in boundary predictions.

There lacks description of training details.

[ref1] Rethinking Semantic Segmentation: A Prototype View. CVPR 2022.

=======

overall, I am slightly positive to the article. However, since I am not an expert in casual inference, I will see other reviewers' comments and make the final decision.

**Questions:**

see [weaknesses]

---

> ### Author Response · Authors · 2023-11-14
>
> **Q1. It is unclear how Eq. 4 is derived. Since this is probably the most important part of the algorithm, more explanations should be given.**
>
> **A1.** We would like to remind our writings ***"we proceed to enhance the likelihood $p(Y |t', m)$ for effectively clustering pixel-level semantics. ... However, we cannot directly compute this likelihood as in standard supervised learning, primarily because there are no available pixel annotations. Instead, we substitute the likelihood of unsupervised dense prediction to concept-wise self-supervised learning based on Noise-Contrastive Estimation (NCE) [R1]"***. in the paragraph of Concept-wise Self-supervised Learning at Sec 3.2.
>
> In unsupervised setup, we adopt common contrastive learning for grouping pixel representation, which we have verified that this learning is closely related with the causal inference. In other words, we borrow the formulation of NCE to build the likelihood $p(Y |t', m)$ on our purpose: unsupervised dense prediction without any annotations, therefore its derivation is fundamentally based on empirical study of NCE [R1]. Here, $t'$ and $m$ are used to find positive $y^+$ and negative $y^-$ features for the given anchor features $y$. Please see further explanation in Answer 2.
>
> ---
>
> **Q2. I am confused about how "find patch feature points in satisfying $\mathcal{D}_M[\text{id}_q, :]$" is implemented in practice. As far as I understand, only summarizes prototype-prototype similarities; then how to select features based on the aforementioned rule?**
>
> **A2.** We first kindly remind that we have included comprehensive explanations of feature selections in Appendix B. Here, we add more intuitive explanation for the question. For example, we have a concept book $M$ with the number of four concepts. Then, we can compute the distance matrix $\mathcal{D}_{M} \in \mathbb{R}^{4\times 4}$.
>
> Let us assume that the first row vector of the distance matrix is $\mathcal{D}_{M}[0,:] = [1, 0.02, 0.47, 0.18]$ which represents the concept-wise distance vector between the first concept (index=$0$) and all of the four concepts in $M$.
> When we set positive and negative relaxation to $\phi^+ = 0.3$ and $\phi^- = 0.1$, then the third concept (index=$2$) is considered as positive concept to the first concept (index=$0$) due to the value of $0.47>\phi^+$, and the second concept (index=$1$) gets to be negative concept to the first concept (index=$0$) due to the value of $0.02<\phi^-$. Next, once self-supervised DINO feature $T\in\mathbb{R}^{5(h)\times 5(w) \times 768(c)}$ is given and its spatial features $T[0, 0, :], T[2, 4, :], T[4, 1, :] \in\mathbb{R}^{768}$ are the closest to, respectively, the first concept (index=0), the second concept (index=1), and the third concept (index=2), then $T[0, 0, :]$ and $T[2, 4, :]$ are the negative feature to be far away from each other, and $T[0, 0, :]$ and $T[4, 1, :]$ are the positive feature to get clustered. Because DINO features are frozen so that we cannot learn DINO features, we instead deal with the segmentation head output features $y=Y[0, 0, :], y^+=Y[2, 4, :], y^-=Y[4, 1, :]$ to be separated or clustered,  by enhancing the likelihood of semantic groups: $p(y|t', m) = \frac{\exp(\text{cos}(y, y^+)/\tau)}{\exp(\text{cos}(y, y^+)/\tau) + \exp(\text{cos}(y, y^-)/\tau)}$.
>
> ---
>
> **Q3 For the statement: "we set tight negative relaxation ..., ... emphasizing that hard negative mining is crucial to advance self-supervised learning", does a tight negative relaxation indicate easier negative mining?**
>
> **A3.** No. Tight negative relaxation means hard negative mining. We have already mentioned ***"hard negative mining is crucial to advance self-supervised learning"*** with citing many references. Furthermore, as illustrated in Figure 5 (c), we have even already experimented the negative relaxation $\phi^-$ across 0.1 (tight) to 0.7 (smooth) and showed the tight negative relaxation (hard negative mining) is crucial to perform concept-wise self-supervised learning well, either.
>
> ---
>
> **References**
>
> * [R1] Noise-contrastive estimation: A new estimation principle for unnormalized statistical models. In Proceedings of the thirteenth international conference on artificial intelligence and statistics, pp. 297–304. JMLR Workshop and Conference Proceedings, 2010.

---

> ### Author Response · Authors · 2023-11-14
>
> ---
>
> **Q4. The concept prototype bears high similarity with the work [R2]. I am wondering whether the sinkhorn-knopp algorithm used in [R2] can be used here for prototype generation.**
>
> **A4.** It is not feasible to use sinkhorn-knopp algorithm in our work. [R2] generates class-wise numerous prototypes using ***"labeled annotations"!*** based on *supervised training*, where Sinkhorn-Knopp algorithm is employed together with Cross-Entropy Loss (supervised loss) to discriminate which prototypes are matching to the class labels (*i.e.,* many-to-one matching). Here, the reason why Sinkhorn-Knopp is necessary in [R2] is not the purpose of generating prototypes themselves but of increasing stable matching between the generated prototypes and concepts. Instead, generating prototypes is rather related with Cross-Entropy Loss so that without Cross-Entropy Loss, Sinkhorn-Knopp will be not working due to the absence of pixel annotation's actual guidance to the clustering or separating. Instead, we would like to highlight that we have conducted ablation for other prototype generation methods in Table 4.
>
> ---
>
> **Q5. From Fig. 4, it appears that the method works particularly well in object boundaries (very close to ground-truths for, e.g., persons). While this is impressive for unsupervised segmentation, I am curious how the algorithm improves over other methods in boundary predictions.**
>
> **A5.** The primary goal for unsupervised semantic segmentation is grouping to coarse-level semantics for the pixel representation. We achieved SOTA results for USS task, and it is mainly due to ***mediator*** for explicitly quantized concepts based on the causal perspective. By comparing concept-wise distances for each explicit concepts, the boundary pixels can be considered more noticeable than inner pixel regions (high concept distance in boundary), which is machted with our purpose of course-level grouping.
>
> In addition, we also concurrently remind that we use common post-processing techniques such as Fully-Connected CRF [R3]. The sharpening is mainly attribute to such processing. It is worth to note that previous works: STEGO, HP, and ReCo+ equally have used CRF for a post-processing, of which original paper's Figure 1(e) shows significant boundary segmentation performance between "tree leafs". This CRF algorithm is commonly used in segmentation area such as DeepLab [R4] to improve qualitative results, thus we would like to note that the noticable quality in boundary is attributed to the CRF post processing. Please see the ablation results for the existence of CRF in Table 4.
>
> ---
>
> **Q6. There lacks description of training details.**
>
> **A6.** In Appendix B, we have already described a concrete implementation for training details of CAUSE about (B.1) Maximizing Modularity, (B.2) Transformer-based Segmentation Head, (B.3) Anchor Selection, (B.4) Positive \& Negative Concept Selection, (B.5) Concept Bank, and (B.6) Image Resolution and Augmentation.
>
> Though, based on Answer 1 and 2, once we take potential next steps (cam-ready), we will add the explanation on one-extra page with a simple example to make readers more easily understandable to the training details of concept-wise self-supervised learning and selecting positive and negative concept in Sec 3.3.
>
> ---
>
> **References**
>
> * [R2] Rethinking semantic segmentation: A prototype view. Proceedings of the IEEE/CVF Conference on Computer Vision and Pattern Recognition. 2022.
>
> * [R3] Efficient inference in fully connected crfs with gaussian edge potentials. Advances in neural information processing systems 24 (2011).
>
> * [R4] Deeplab: Semantic image segmentation with deep convolutional nets, atrous convolution, and fully connected crfs IEEE transactions on pattern analysis and machine intelligence 40.4 (2017): 834-848.

---

> > ### Author Response · Authors · 2023-11-21
> > **Remind Comment**
> >
> > We kindly remind the reviewer that the discussion period will conclude in the next 24 hours.
> >
> > We have comprehensivly answered to the raised questions, and we believe that many of them can be easily addressed with further explanation, some of which are already included in the Appendix.
> >
> > We are looking forward to further discussions and any additional feedbacks.

---

### Author Response · Authors · 2023-11-14
**Thank you for feedbacks!**

We would like to appreciate to reviewers for their valuable reviews. We will incorporate the feedbacks into the potential final version.

We thank to *Reviwer ivD5* and *Reviwer vbSV* for acknowledging our novelty on causal perspective in unsupervised domains. We also appreciate for recognizing our state-of-the-art performance in unsupervised semantic segmantation for *Reviwer 1Vow*, *Reviwer zoBW*, *Reviwer ivD5*.

We would like to improve the quality of our paper during the active disccusion period and contribute to potential readers and ICLR community.

---

### Author Response · Authors · 2023-11-17

We would like to kindly remind the reviewers that the discussion period is scheduled to end on November 22 (UTC-12h).

Your additional feedbacks during this discussion period is highly appreciated. We look forward to incorporating any further suggestions you may have to enhance the contribution of our paper and address any potential misunderstandings.

---

### Meta-Review · Area_Chair_owbF · 2023-12-06

**Metareview:**

This paper introduces CAUSE, a novel framework for unsupervised semantic segmentation. Leveraging insights from causal inference, CAUSE utilizes an intervention-oriented approach to define two-step tasks for unsupervised prediction. It constructs a concept clusterbook as a mediator in the first step, representing concept prototypes at different granularity levels. The mediator links to subsequent concept-wise self-supervised learning for pixel-level grouping.

While the idea of connecting the unsupervised segmentation problem with causal inference is novel and interesting, the formulation of the problem lacks rigour. Firstly, the random variables in the causal graph are not well justified. Especially, the confounder variable U, what is the possible distribution of U? From the explanation in the paper, U does not look like a random variable that is correlated with T and Y. When constructing a causal graph, it is essential to check whether the data generating process implied by the graph makes sense. Second, there is a gap between front door adjustment and the proposed method. It is unknown how to derive the proposed method from the front door formula.

The method proposed in this paper is not necessarily a causal method. I would suggest the authors either further strengthen the connection between causal inference in a more principled manner or present the proposed method from different perspectives if a rigorous connection between causal inference and the proposed method cannot be established.

**Justification For Why Not Higher Score:**

The formulation of the problem is ad-hoc. A more rigorous treatment of the problem from a causal perspective is needed.

**Justification For Why Not Lower Score:**

N/A

---

### Decision · Program_Chairs · 2024-01-16

Reject